# GUMICS-4 analysis of ICME impact at Earth during low and typical Mach number solar wind

Antti Lakka[1], Tuija I. Pulkkinen[1,6], Andrew P. Dimmock[2], Emilia Kilpua[3], Matti Ala-Lahti[3], Ilja Honkonen[4], Minna Palmroth[3], and Osku Raukunen[5]

[1]Department of Electronics and Nanoengineering, Aalto University, Finland
[2]Swedish Institute of Space Physics, Uppsala, Sweden
[3]Department of Physics, University of Helsinki, Helsinki, Finland
[4]Finnish Meteorological Institute, Helsinki, Finland
[5]Department of Physics and Astronomy, University of Turku, Turku, Finland
[6]University of Michigan, Ann Arbor, USA

**Correspondence:** Antti Lakka (antti.lakka@aalto.fi)

**Abstract.**

We study the response of the Earth's magnetosphere to fluctuating solar wind conditions during interplanetary coronal mass ejections (ICME) using the Grand Unified Magnetosphere-Ionosphere Coupling Simulation (GUMICS-4). The two ICME events occurred on 15–16 July 2012 and 29–30 April 2014. During the strong 2012 event, the solar wind upstream values reached up to 35 particles/$cm^3$, speed to 694 km/s, and interplanetary magnetic field to 22 nT, giving a Mach number of 2.3. The 2014 event was a moderate one, with the corresponding upstream values of 30 particles/$cm^3$, 320 km/s and 10 nT, indicating Mach number of 5.8. We examine how the Earth's space environment dynamics evolves during both ICME events from both global and local perspectives, using well-established empirical models and in-situ measurements as references. We show that in the large scale, and during moderate driving, the GUMICS-4 results are in good agreement with the reference values. However, the local values, especially during high driving, show more variation: Such extreme conditions do not reproduce local measurements made deep inside the magnetosphere. The same appeared to be true when the event was run with another global simulation. The cross-polar cap potential (CPCP) saturation is shown to depend on the Alfvé n Mach number of the upstream solar wind. However, care must be taken in interpreting these results, as the CPCP is also sensitive to the simulation resolution.

## 1 Introduction

Present understanding is that the coupling of the solar wind and the Earth's magnetosphere occurs via magnetic reconnection (Dungey, 1961) and viscous processes (Axford and Hines, 1961) such as the Kelvin-Helmholtz instability (e.g. Nykyri and Otto (2001)) and diffusion (Johnson and Cheng, 1997). Although viscous processes may play a strong role, particularly when the interplanetary magnetic field (IMF) is northward (IMF $B_Z > 0$ nT) (e.g. Osmane et al. (2015)), magnetic reconnection on the dayside magnetopause is responsible for the majority of plasma transport across the magnetopause during southward interplanetary magnetic field IMF (IMF $B_Z < 0$ nT), allowing the solar wind to drive activity in the Earth's space environment

(Nishida, 1968; Koustov et al., 2009). The intervals of extended periods of strongly southward IMF typically arise when the Earth encounters an interplanetary coronal mass ejection (ICME) (see e.g. Kilpua et al. (2017b)). ICMEs are interplanetary counterparts of coronal mass ejections (CMEs), large eruptions of plasma and magnetic field from the Sun, driving the strongest geomagnetic disturbances (e.g., Gosling et al. (1991); Huttunen et al. (2002); Richardson and Cane (2012); Kilpua et al. (2017a)). The signatures of ICMEs at 1 AU include high helium abundance (Hirshberg et al., 1972), high magnetic field magnitude and low plasma beta (Hirshberg and Colburn, 1969; Burlaga et al., 1981), low ion temperatures (Gosling et al., 1973), and smooth rotation of the magnetic field (Burlaga et al., 1981). While there have been attempts to form a universal set of signatures to describe ICMEs (Gosling, 1990; Richardson and Cane, 2003), they vary significantly such that no single set of criteria are able to describe all the ICME events, and none of them are unique to ICMEs. For example, only one third to one half of all the ICMEs have a magnetic flux rope (or a magnetic cloud) (e.g. Gosling, 1990; Richardson and Cane, 2003), whose signatures combine enhanced magnetic field, reduced proton temperature, and the smooth rotation of the magnetic field over an interval of a day (Burlaga et al., 1981). While magnetic clouds are the most studied part of ICMEs due to their significant potential to cause large space storms, their relationship to the entire ICME sequence still pose many questions (e.g., Kilpua et al. (2013)). Moreover, if the ICME is sufficiently faster than the ambient solar wind plasma, a shock is formed ahead of the ICME (Goldstein et al., 1998), with a region of compressed solar wind plasma between the leading shock front and the magnetic cloud, referred to as the sheath region.

The sheath and ejecta are the most distinctive parts of ICMEs (see e.g. Kilpua et al. (2017b)), and both can drive intense magnetic storms (e.g. Tsurutani et al. (1988); Huttunen and Koskinen (2004)). However, they have clear differences in their solar wind conditions and consequently, their coupling to the magnetosphere is different (Jianpeng et al., 2010; Pulkkinen et al., 2007; Kilpua et al., 2017b). ICME sheaths typically include high solar wind dynamic pressure and fluctuating IMF, including both northward and southward orientations within a short time period (Kilpua et al., 2017b). The duration of the sheath is also typically shorter than the following cloud, for example Zhang et al. (2012) obtained the average values of 10.6 and 30.6 hours for sheaths and clouds, respectively. Sheaths are known to enhance high-latitude ionospheric currents (Huttunen and Koskinen, 2004), and they are found to have higher coupling efficiency than clouds (Yermolaev et al., 2012). The clouds typically enhance the equatorial ring current (Huttunen and Koskinen, 2004).

Due to potential for strongly southward IMF orientation, ICME magnetic clouds drive enhanced magnetospheric activity. Moreover, during cloud events, due to the combination of generally high magnetic fields and low plasma densities, the solar wind Alfvén Mach number $M_A$ can reach quite low values and even be close to unity. The role of $M_A$ for solar wind - magnetosphere coupling has been highlighted in recent studies (Lavraud and Borovsky, 2008; Lopez et al., 2010; Myllys et al., 2016, 2017). In particular, the role low $M_A$ conditions typical to ICME magnetic cloud for the saturation of the ionospheric cross-polar cap potential CPCP has been a subject of several studies (e.g. Ridley, 2005, 2007; Lopez et al., 2010; Wilder et al., 2015; Myllys et al., 2016; Lakka et al., 2018).

Global MHD models have been used to study the effects of ICMEs on the magnetospheric and ionospheric dynamics. Wu et al. (2015) used the H3DMHD model (e.g. Wu et al., 2007) to examine a CME event on March 15, 2013. They found that the high-energy solar energetic proton time-intensity profile can be explained by the interaction of a CME-driven shock with the

heliospheric current sheet embedded within nonuniform solar wind. A recent paper by Kubota et al. (2017) studied the Bastille Day geomagnetic storm event (July 15, 2000) driven by a halo CME. They found that the inclusion of auroral conductivity in the ionospheric part of the global MHD model by Tanaka (1994) led to saturation of the CPCP without any effect on the field-aligned currents, thus suggesting a current system with a dynamo in the magnetosphere and a load in the ionosphere.

The difficulty in assessing these studies is that they often do not include uncertainty estimate of the model results, while the methods are different for each study. Moreover, while the different MHD simulations are based on the same plasma theory, the approaches are different in terms of exact form of the equations, the numerical solutions, and the initial and boundary conditions, thus making comparisons of different models difficult. Nonetheless, understanding of the performance limits of the simulations is essential for meaningful comparisons to in-situ measurements.

Regardless of the different approaches used in gobal codes, the performance of the models have been assessed in several studies. Usually such assessments have been done through comparisons of the simulation results with in situ or remote observations of dynamic events or plasma processes (Birn et al., 2001; Pulkkinen et al., 2011; Honkonen et al., 2013). This is often not easy, as even small errors in the simulation configuration may create large differences with respect to the observations locally at a single point (Lakka et al., 2017), even if the simulation would reproduce the large-scale dynamic sequence correctly. Moreover, recent

studies (Juusola et al., 2014; Gordeev et al., 2015) have shown that none of the codes emerges as clearly superior to the others, each having their strengths and weaknesses. In the absence of uniform code performance testing methodology, validating the results individually is important.

In this study we use the GUMICS-4 (Janhunen et al., 2012), global MHD simulation, and consider two ICME events, one having a significantly stronger solar wind driver than the other. To compare the two events, we use variables that are both

particularly sensitive to upstream changes and used extensively in previous studies, and examine how those variables are affected by the two events. The comparisons include the subsolar magnetopause position, the amount of energy transferred from the solar wind into the magnetosphere, the CPCP, and the magnetic field magnitude within the inner part of the magnetosphere, thus including both global and local variables. We especially focus on periods within the magnetic clouds within the ICMEs, by using two different spatial resolutions. We provide an uncertainty estimate (standard deviation and in some cases also relative

difference) for each quantity by comparing simulation results to well-established references, which include the Shue model (magnetopause location), the epsilon parameter (energy transferred through the magnetopause), the PCI index (CPCP), and in-situ measurements by Geotail and Cluster spacecraft (magnetic field magnitude). Both uncertainty estimate methods are assessed and they are used if the method is valid for the chosen quantity.

This paper is structured in a following way: Section 2 describes GUMICS-4 global MHD code and the simulation setup,

Section 3 describes characteristics of the two ICME events and the executed simulations, Section 4 presents the main results and Section 5 includes the discussion followed by conclusions.

## 2 Methodology

### 2.1 GUMICS-4 Global MHD Simulation

The simulations were executed using the fourth edition of the Grand-Unified Magnetosphere-Ionosphere Coupling Simulation (GUMICS-4), in which a 3D MHD magnetosphere is coupled with a spherical electrostatic ionosphere (Janhunen et al., 2012). The finite volume MHD solver solves the ideal MHD equations with the separation of the magnetic field to a curl-free (dipole) component and divergent-free component created by currents external to the Earth ($\mathbf{B} = \mathbf{B_0} + \mathbf{B_1}(t)$) (Tanaka, 1994). The MHD simulation box has dimensions of 32 ... -224 $R_E$ in $X_{GSE}$ direcion and -64 ... +64 $R_E$ in both $Y_{GSE}$ and $Z_{GSE}$ directions, while the inner boundary is spherical with a radius of 3.7 $R_E$. GUMICS-4 uses temporal subcycling and adaptive cartesian octogrid to improve temporal and spatial resolution in key regions, which means that it only runs on a single processor due to difficulties in parallelizing computations with two adaptive grids. The temporal subcycling reduces the number of MHD computations an order of magnitude while maintaining the local Courant-Friedrichs-Levy (CFL) constraint (J.L. Lions, 2000, p. 121 — 151). The adaptive grid ensures that whenever there are large gradients, the grid is refined thus resolving smaller-scale features especially close to boundaries and current sheets.

The ionospheric grid is triangular and densest in the auroral oval, while in the polar caps the grid is still rather dense, with about 180 km and 360 km spacing used in the two regions, respectively. The ionosphere is driven by field-aligned currents and electron precipitation from the magnetosphere as well as by solar EUV ionisation. Field-aligned currents contribute to the cross-polar cap potential through

$$\nabla \cdot \mathbf{J} = \nabla \cdot [\Sigma \cdot (-\nabla\phi + V_n \times \mathbf{B})] = -j_{||}\left(\hat{\mathbf{b}} \cdot \hat{\mathbf{r}}\right), \tag{1}$$

where $\mathbf{J}$ is current density, $\Sigma$ is the height-integrated conductivity tensor, $\phi$ is the ionospheric potential, $V_n$ the neutral wind caused by the Earth's rotation, $j_{||}$ is the field-aligned current, and $\left(\hat{\mathbf{b}} \cdot \hat{\mathbf{r}}\right)$ is the cosine of the angle between the magnetic field direction $\hat{\mathbf{b}}$ and the radial direction $\hat{\mathbf{r}}$ (Janhunen et al., 2012). Electron precipitation and solar EUV ionisation have contributions on the height-integrated Pedersen and Hall conductivities with solar EUV ionisation parametrized by the 10.7 cm solar radio flux that has a numerical value of $100 \times 10^{-22}$ W/m$^2$. Electron precipitation affects the altitude-resolved ionospheric electron densities, and are used when computing the height-integrated Pedersen and Hall conductivities. The details on the ionsopheric part of GUMICS-4 can be found in Janhunen and Huuskonen (1993) and Janhunen (1996).

The region between the MHD magnetosphere and the electrostatic spherical ionosphere is a passive medium where no currents flow perpendicular to the magnetic field. The magnetosphere is coupled to the ionosphere using dipole mapping of the field-aligned current pattern and the electron precipitation from the magnetosphere to the ionosphere and the electric potential from the ionosphere to the magnetosphere. This feedback loop is updated every 4 seconds.

### 2.2 GUMICS simulations of two ICME events

We use both 0.5 and 0.25 $R_E$ maximum spatial resolutions as well as varying dipole tilt angle in this study. Two complete ICME periods were simulated using 0.5 $R_E$ resolution by starting with nominal solar wind conditions preceding the events,

and ending with nominal conditions following the events. To give GUMICS-4 magnetosphere time to form (Lakka et al., 2017), the simulations were initialized with two hours of constant solar wind driving using upstream values equal to those during the first minute of the actual simulation ($n, |V|, |B|$ values of $4 \text{ cm}^{-3}$, $310 \text{ km/s}$ and $1.1 \text{ nT}$ for the 2012 event, and $11 \text{ cm}^{-3}$, $300 \text{ km/s}$ and $1.8 \text{ nT}$ for the 2014 event).

Due to computational limitations, using the best maximum spatial resolution ($0.25 \text{ R}_\text{E}$) covering both ICME events with full length is not feasible due to long simulation physical time (up to 3.5 days) and resulting long simulation running times. Hence, two additional runs were performed with $0.25 \text{ R}_\text{E}$ maximum spatial resolution in order to gain a more detailed view of the dynamics of the magnetosphere and ionosphere when the ICME magnetic cloud was propagating past the Earth. These runs lasted 6 hours each, and were executed by restarting the $0.5 \text{ R}_\text{E}$ runs with enhanced resolution. Table 1 summarizes all four

simulation runs related to the study.

## 3   Observations of two ICME events

We use the solar wind data from the NASA OMNIWeb service (http://omniweb.gsfc.nasa.gov) and the solar energetic particle data from the NOAA NCEI Space Weather data access (https://www.ngdc.noaa.gov/stp/satellite/goes/index.html). Onset times for the ICME sheath (i.e., the shock time) and the magnetic cloud boundary times are retrieved from the Wind spacecraft

ICME catalogue (https://wind.nasa.gov/ICMEindex.php). Figures 1 and 2 show the upstream parameters during both events. For both figures, IMF $X, Y, Z$ components and the IMF magnitude are shown in panel a, upstream plasma flow velocity $X, Y, Z$ components in panel b, the upstream plasma number density in panel c, upstream Alfvén Mach number (in logarithmic scale) in panel d, energetic proton fluxes for three GOES-15 energy channels between 8–80 MeV in panel e, and the cross-polar cap potential from the GUMICS-4 simulation in panel f. Figure 1 includes time range from 09:00 UT, July 14 to 15:00 UT, July

17, 2012, while Figure 2 shows the period from 19:00 UT, April 28 to 17:00 UT, May 1, 2014. The time of the ICME shock, and the start and end times of the ICME are marked with vertical red lines in both figures. The grey-shaded regions indicate the time periods simulated with the maximal $0.25 \text{ R}_\text{E}$ spatial resolution. Both IMF and plasma flow velocity components are given in GSE coordinate system, which is also the coordinate system used by the GUMICS-4 simulation.

Figure 1 shows the arrival of the leading shock at 18:53 UT on July 14, 2012 as the simultaneous abrupt jump in the plasma and

magnetic field parameters and the following ICME sheath as irregular directional changes of the IMF and compressed plasma and field. The energetic particle fluxes for the two lower energy channels increase until after the shock passage, which suggests continual particle acceleration in the shock driven by the ICME. At 06:54 UT on July 15, the onset of the ICME magnetic cloud is identified by strong southward turning of the IMF. Significant reduction in the number density, and the clear decrease in the variability of the interplanetary magnetic field. During the next 45 hours, the IMF direction stayed strongly southward

while slowly rotating towards less southward orientation. We note that in the trailing part of the ICME, the field changes rather sharply to northward, thereafter continuing to rotate southward again. We cannot rule out that this end part is not another small ICME, but as our study focuses on the strong southward magnetic fields in the main part of the ICME we do not consider the origin of this end part further here.

The ICME on April 2014 was slower than the July 2012 ICME and its speed was very close to the ambient solar wind speed. Hence, no shock, nor clear sheath developed ahead of this ICME. The onset of the ICME-related disturbance is marked by the increased plasma number density followed by a rapid decrease and a clear southward turning of the IMF at 20.38 UT on April 29 (Figure 2). The weaker activity is also evident by the lack of energetic particle fluxes above background in the magnetosphere. The very early phase of this cloud may contain some disturbed solar wind (the region of higher density and fluctuating field), but we do not identify it as a sheath and focus our study on the effects of the cloud proper.

Both magnetic clouds are characterized by low Alfvén Mach number. In the 2012 case, $M_A$ drops even below unity and is 1.9 on average during the cloud structure, while during the 2014 magnetic cloud, the minimum $M_A$ was 3.8 and the average was 5.8.

The 2012 event features generally larger CPCP, with values above 40 kV, and reaching 70 kV (Figure 1f). On the other hand, during the 2014 event the CPCP peaks early at 50 kV and subsequently reduces to 20 kV (Figure 2f). GUMICS-4 CPCP values depend on grid resolution and while lower grid resolution may result as substantially lower CPCP values than the observed values (Gordeev et al., 2015), higher resolution leads to higher CPCP values (e.g. Lakka et al., 2018) and thus better agreement with the observations.

The 2012 ICME event is considerably longer than the 2014 event, with 57h 26min total duration, of which 12h 1min are sheath, and 45h 25min part of the magnetic cloud passage. The 2014 event lasted 21 h 13 min in total. The 2012 ICME had larger effects on magnetospheric activity, as the solar wind driving was considerably stronger, with the average IMF magnitude and solar wind speed of 14 nT and 490 km/s, respectively, compared with 8.5 nT and 303 km/s of the 2014 event. The maximum IMF magnitude and upstream solar wind speed were also larger during the 2012 event, with 21 (10) nT and 660 (321) km/s maximum values measured during the 2012 (2014) cloud. However, while maximum number density was higher during the 2012 magnetic cloud (36 cm$^{-3}$ vs. 30 cm$^{-3}$), the average number density was considerably higher during the 2014 event (2012: 2 cm$^{-3}$ vs. 2014: 12 cm$^{-3}$).

During the two ICME events, data from the Cluster 1 (hereafter Cluster) and Geotail satellites were available from the CDAWeb service (https://cdaweb.sci.gsfc.nasa.gov/index.html/). Figure 3 shows the orbits of Cluster (blue) and Geotail (green) along with the magnetopause location (black) from the empirical Shue model (Shue et al., 1997) on the $XY$ plane (figures 3a and 3c) and on the $XZ$ plane (figures 3b and 3d) for both events. The magnetopause position is computed for the most earthward magnetopause location during the events, while the orbit tracks include intervals of nominal upstream conditions before and after the ICME events. Start and end points of the time intervals are marked with a cross and a triangle, respectively. Dots mark the points where satellite orbits intersect (located visually) the innermost position of the magnetopause. The variability of the magnetopause position means that between those orbit tracks the S/C may cross to outside the magnetosphere. The used coordinate system is GSE. Based on figure 3, the Cluster spacecraft orbits inside of the magnetosphere throughout the 2012 event and for most of the 2014 event. On the other hand, Geotail is outside the magnetosphere an extended period during July 16-17, 2012 as well as during several periods in April–May 2014.

Figures 4 and 5 show time series of the magnetic field magnitude $|B|$ along the Geotail (panel a) and Cluster (panel b) orbits during the 2012 and 2014 events. Green (Geotail) and blue (Cluster) curves show the observations, while the black (magenta)

curve shows the magnetic field magnitude along the spacecraft orbits in GUMICS-4 simulation using 0.5 (0.25) $R_E$ maximum spatial resolution. The yellow-shaded regions in panels a and b indicate times when the spacecraft may encounter magnetopause crossings. Note that a logarithmic scale is used for the Cluster data. Panel c in both figures shows the radial distance of the spacecraft from the center of the Earth. Note that satellite measurements have been interpolated over long (several hours) datagaps, most notably on July 16, 12:15–18:45 UT.

At the start of the 2012 event, Geotail resides in the plasma sheet, but quickly moves to the boundary layer (roughly July 14, 16:00 UT to July 15, 06:00 UT), after which it enters the lobe as the cloud proper hits the magnetosphere. At around the end of the data gap at the end of July 16, the spacecraft moves to the low latitude boundary layer and the magnetosheath (identified from plasma data not shown here).

At the start of the 2012 event, Cluster is near perigee recording field values dominated by the dipole contribution. Cluster exits the ring current region around 16:00 UT on July 14, and enters the plasma sheet. A brief encounter in the lobe is recorded between roughly 18:00 UT July 15 and 06:00 July 16. A second period in the inner magnetosphere commences around 12:00 UT on July 16, with exit to the lobe after 00:00 UT July 17 (identified from plasma and energetic particle data not shown here).

## 4 Analysis

### 4.1 Global dynamics

Figures 6 and 7 show the effect of upstream IMF $B_Z$ (panel a), and solar wind dynamic pressure (panel b) on the magnetopause nose (panel c), total energy through the dayside magnetopause nose position (panel d) and the ionospheric cross-polar cap potential CPCP (panel e) during the simulated intervals shown in figures 1 and 2. The 0.5 $R_E$ resolution run results are shown in black, and 0.25 $R_E$ resolution results are shown in magenta. Grey shaded area highlights the 6-hour interval simulated using both resolutions. Blue and green curves indicate reference values (see below) and solar wind upstream conditions, respectively. As a metrics for validating the simulation results, we use the magnitude of the relative difference (given as $\delta$ in panel c of figures 6 and 7)

$$\delta = \left| \frac{x_{\text{ref}} - x_{\text{GUMICS}-4}}{x_{\text{ref}}} \right|, \tag{2}$$

in which $x$ is the GUMICS-4 variable and $x_{ref}$ refers to the reference parameter value of the variable. An average $\delta$ value is computed for each ICME simulation phase (nominal solar wind, sheath, cloud) for both 0.5 $R_E$ and 0.25 $R_E$ resolution runs. These percentage values can be found in table 2. We also compute standard deviation (SD) for the reference vs. GUMICS-4 results. A single SD value (given in panels c, d and e) is computed for the 0.5 $R_E$ resolution runs to illustrate how similar the temporal evolution is over time scales of days for GUMICS-4 and the reference parameter.

Figures 6a and 6b show that the IMF $B_Z$ fluctuates approximately between -5...+5 nT during nominal solar wind conditions, while the solar wind dynamic pressure is steady and low. At the onset of ICME sheath, both $B_Z$ and dynamic pressure start fluctuating with increased amplitude. Moreover, after the onset of ICME cloud, the orientation of the IMF slowly rotates from southward to northward with the solar wind dynamic pressure decreasing rapidly and remaining low until the end of the

simulated interval. This behaviour is somewhat similar during the 2014 event (figures 7a–7b), with the exception of missing high amplitude fluctuations due to absence of a distinct ICME sheath.

In GUMICS-4, we identify the magnetopause nose position as a single grid point having the maximum value of $J_Y$ along the Sun-Earth line, using one-minute temporal resolution, smoothed using 10-min sliding averages. This value is compared with the Shue (Shue et al., 1997) empirical magnetopause model. For simplicity, the nose of the magnetopause is referred to a magnetopause. Figure 6c shows that at the onset of ICME sheath, the magnetopause moves Earthward as a consequence of changing upstream conditions, which is followed by Sunward return motion lasting until the end of the ICME event. The average $\delta$ is highest during the cloud (8%) and lowest (2.5%) during nominal solar wind conditions. During ICME sheath, average $\delta$ is 4.5%. During the 2014 event, the magnetopause starts moving Earthward at least 10 hours before the onset of ICME cloud (figure 7c), as the dynamic pressure increases, with IMF $B_Z$ staying positive. After the onset however, the magnetopause moves Sunward for a few hours until slowly moving Earthward again. The difference in average $\delta$ between cloud and nominal solar wind conditions is lower than for the 2012 event, as the respective values are 3.3% and 2.4%.

The grey-shaded region in figure 6c shows that during the first four hours of the 6-hour run the magnetopause position predictions (black and magenta curves) by GUMICS-4 are within 5% of the Shue et al. (1997) model (blue curve). During the last 2 hours, however, there are more fluctuations in the GUMICS-4 magnetopause position, especially in the 0.5 $R_E$ resolution run. From July 15, 21:00 UT to July 16, 01:00 UT the simulation runs agree on the magnetopause location and also with the Shue model, with differences within 10% all the time of the first 4 hours. However, the last two hours show more variations between the three curves: The finest resolution show slight outward motion of the magnetopause, which toward the end of the period is less than that predicted by the Shue model. On the other hand, the 0.5 $R_E$ resolution run shows inward indentations followed by outward motion consistent with the Shue model. Overall, the 0.5 $R_E$ resolution run is 58% of the time within 10% of the Shue model, and the 0.25 $R_E$ resolution run agree 67% of the time within 10% of the Shue model. Despite the fact that average relative difference is slightly lower for the 0.5 $R_E$ resolution run (4.9%) than for the 0.25 $R_E$ resolution run (5.6%), over the entire 6-hour periods, the 0.25 $R_E$ run is within 10% of the Shue model 92% of the time, while the 0.5 $R_E$ run reaches within 10% 89% of the time due to the 0.5 $R_E$ run being more inclined toward moving more Earthward during the last two hours of the 6-hour period.

The time evolution of the magnetopause position during the 6-hour period in Figure 7 is similar for both spatial resolutions, with both simulation runs responding similarly to small upstream fluctuations. Both simulation runs stay within 10% of the Shue model prediction for the entire 6-hour period. Average relative difference is only slightly lower for the higher resolution run (3.2%), than for the lower resolution run (4.5%).

Overall, the higher-resolution run yielded better agreement with the magnetopause location especially for a moving magnetopause nose (2012 event), because increasing the spatial resolution sharpens the gradients and allows better identification of the location of the maxima (Janhunen et al., 2012). Comparison of the runs shows, however, that the results are consistent with each other, indicating that the lower-resolution run is providing similar large-scale dynamics as the finer-resolution run. Furthermore, increased $\delta$ during the 2012 ICME cloud and overall higher $\delta$ during the 2012 event indicate that GUMICS-4

accuracy in the magnetopause nose position prediction is better during weaker solar wind driving. This is further demonstrated by the standard deviation values, which are 0.661 for the 2012 event, and 0.321 for the 2014 event (see figures 6c and 7c).

Total energy through the dayside magnetopause is computed by evaluating the energy flux incident at the (Shue) magnetopause, and it is evaluated from

$$\mathbf{K} = \left( u + p - \frac{B^2}{2\mu_0} \right) \mathbf{V} + \frac{1}{\mu_0} \mathbf{E} \times \mathbf{B}, \tag{3}$$

where $u$ is the total energy density, $p$ pressure, $B$ magnetic field, $\mathbf{V}$ flow velocity and $\mathbf{E} \times \mathbf{B}$ the Poynting flux, and its component perpendicular to the magnetopause surface. As is shown in figure 6c, the relative difference magnitude $\delta$ in the magnetopause nose location can reach up to 30% values. To avoid underestimating the size of the magnetosphere, we evaluate the magnetopause surface by moving the radial distance of each Shue magnetopause surface value 30% further away from the

Earth. This surface is then used in integrating the energy flux values entering the magnetosphere Sunward of the terminator ($X > 0$ $R_E$). The results are shown for the 2012 event in figure 6d for both 0.5 and 0.25 $R_E$ resolution runs along with the computed $\epsilon$-parameter (Perreault and Akasofu, 1978):

$$\epsilon = \frac{4\pi}{\mu_0} V B^2 \sin^4(\frac{\theta}{2}) l_0^2, \tag{4}$$

where $\mu_0$ is vacuum permeability, $B$ and $V$ are the magnitudes of the IMF and solar wind plasma flow velocity, $\theta$ is the IMF

clock angle, and $l_0$ is an empirically determined scale length.

While both resolution runs agree with each other, it is evident that their numerical values are quite far from the reference, $\epsilon$-parameter. It should be noted however, that the $\epsilon$-parameter is not scaled to represent the energy input, but the energy dissipated in the inner magnetosphere (Akasofu, 1981). Thus the relative difference is not a good metrics to describe the difference between GUMICS-4 and the $\epsilon$-parameter and thus we are not using it in this paper. However, general temporal evolution is

similar for most parts of ICME cloud, with both GUMICS-4 and the $\epsilon$-parameter reproducing steep increase at the onset of cloud as well as subsequent slow decrease, as is shown by the computed SD value in figure 6d (2.263). As in the case of the 2012 event, the two simulation runs using different spatial resolutions are almost inseparable in terms of the incoming solar wind energy during the 2014 event (Figure 7d). During moderate solar wind driving in 2014, GUMICS-4 is closer to the $\epsilon$-parameter, with considerably lower SD value (0.725) compared with the 2012 event. This is an interesting characteristics of

the $\epsilon$-parameter warranting further study.

Differences between the simulations executed using different spatial resolutions in local measures, such as the magnetopause nose position, do not show in global variables, such as the total energy through the dayside magnetopause surface. As can be seen in Figure 6d, the curves of the two different spatial resolution runs are almost identical. This emphasizes that integrated quantities, such as energy, give a better representation of the true physical properties of the magnetosphere in the GUMICS-

4 solution and are not dependent on grid resolution (Janhunen et al., 2012). We acknowledge that using more sophisticated methods for identifying the magnetopause surface from the simulation could potentially lead to some changes in the results. The Shue model was used for its simplicity and computational ease. Our results agree in general with Palmroth et al. (2003) who identified the magnetopause by using plasma flow streamlines from GUMICS-4, indicating that the use of the Shue model is not introducing large errors in the energy estimates.

The magnetosphere – ionosphere coupling, here illustrated by the CPCP time evolution in Figure 6e, is compared with the polar cap index (Ridley and Kihn, 2004) computed as

$$PCI = 29.28 - 3.31 sin(T + 1.49) + 17.81 PCN,  \tag{5}$$

where $T$ is month of the year normalized to $2\pi$, and $PCN$ is the nothern polar cap index retrieved from OMNIWeb. The PCI is a very indirect proxy (based on a single-point measurement only) for the CPCP, and thus the comparisons must be interpreted with great care. Also, taking into account that one of the well-known feature of GUMICS-4 is lower predicted CPCP values compared with its contemporaries (Gordeev et al., 2015), it is of little importance to report the relative differences in CPCP values with the PCI index as a reference. However, in terms of the SD values, GUMICS-4 and the PCI index show better agreement in the temporal evolution of CPCP during the 2014 event (SD = 15.838) than during the 2014 event (SD = 5.107). It is apparent that these SD values are clearly highest of all three (magnetopause nose, energy, CPCP) for both events. This is in part due to the ionospheric (local) processes contributing to the PCI index but not related to the large-scale potential evolution.

## 4.2 Saturation of the Cross-polar cap Potential

Figures 8 and 9 show the CPCP (both northern and southern hemispheres) as a function of the solar wind electric field $E_Y$ component for both ICME events. Color-coding marks the IMF magnitude in figures 8a and 9a, solar wind speed in figures 8b and 9b, and the upstream Alfvén Mach number in figures 8c and 9c. Every data point in Figure 8 (9) is computed from 10-minute averages, binned by $E_Y$ with 1.0 (0.5) mV/m intervals. The ICME sheath (solid circles) and cloud (solid squares) periods as well as the nominal solar wind conditions (solid triangles) prior to and following the events are analyzed separately. Note that here only the coarse grid (0.5 $R_E$) simulation results are used, as we analyze the effects during the entire magnetic cloud and sheath periods including times before and after the event not covered by the high-resolution run.

Figure 8 shows that the response of the CPCP to the upstream $E_Y$ is quite linear during the magnetic cloud (squares) when solar wind driving electric field $E_Y$ is below 5 mV/m, during nominal solar wind conditions (triangles), and ICME sheath (diamonds). However, the polar cap potential first decreases and subsequently saturates during the cloud when the solar wind driving is stronger ($E_Y > 5$ mV/m). For the 2012 event, we refer to the $E_Y$ range from 0 to 5 mV/m as the linear regime, and from 5 mV/m upward as the non-linear regime.

Figure 8a shows the obvious result that highest $E_Y$ values are associated with highest IMF magnitudes. However, it also shows that the largest IMF magnitudes are associated with the non-linear regime, indicating that strong upstream driving leads to CPCP saturation. In addition, Figure 8b suggests that the increase of the CPCP in the linear regime is clearly higher for lower velocity values (cloud structure), than for higher velocity values (sheath and nominal conditions). Generally, this agrees with the previous studies utilizing statistical (Newell et al., 2008) and numerical (Lopez et al., 2010) tools. The latter authors suggest that this is caused by the solar wind flow diversion in the pressure gradient-dominated magnetosheath; faster solar wind will produce more rapid diversion of the flow around the magnetosphere, and thus smaller amount of plasma will reach the magnetic reconnection site.

Figure 8c shows that the upstream Alfvén Mach number $M_A$ is at or above 4 ($M_A \geq 4$) during the nominal solar wind conditions and during the ICME sheath, while during the magnetic cloud $M_A$ resided below 4 and almost reaches unity. This supports the interpretation that saturation of the CPCP depends on the upstream Alfvén Mach number $M_A$ such that saturation occurs only when $M_A$ values fall below 4. The dependence of the CPCP saturation on $M_A$ is well-known, documented both in

measurements (Wilder et al., 2011; Myllys et al., 2016) and in simulation studies (Lopez et al., 2010; Lakka et al., 2018).

Figure 9 agrees with the view presented above, as the response of the CPCP to the upstream $E_Y$ during the 2014 event is quite linear regardless of the IMF magnitude (Figure 9a), plasma flow speed (Figure 9b), or the large-scale solar wind driving structure (ICME cloud or nominal solar wind). This is apparently because solar wind driving is substantially weaker during the 2014 event than during the 2012 event, with the IMF magnitude reaching barely 10 nT, and upstream plasma flow speed

varying only of the order of 10 km/s. As a result, the upstream Alfvén Mach number $M_A > 4$ throughout the ICME event as well as during the nominal solar wind conditions. The high polar cap potential values for the lowest $E_Y$ bin is associated with the large density enhancement driving polar cap potential increase before the arrival of the cloud proper.

Figure 10 shows the region 1 and region 2 field-aligned current (FAC) system coupling the magnetosphere and the ionosphere (e.g. Siscoe et al. (1991)). The four panels show how field-aligned currents are distributed in the northern hemisphere iono-

sphere in July 16, 2012 at 01:00 UT and 03:00 UT at 0.5 $R_E$ maximum resolution (figures 10a–10b) and at 0.25 $R_E$ maximum resolution (figures 10c–10d). Current density is shown both as color coding and contours, while the white dotted line depicts the polar cap boundary. The distribution of the FAC do not change much in either of the simulations, thus suggesting that the coupling of the magnetosphere and the ionosphere remains relatively constant. However, as is shown in figure 6e, the CPCP shows different temporal evolution based on the used spatial resolution, with increasing (constant) CPCP in the 0.5 (0.25)

$R_E$ simulation, thus suggesting that while the magnetosphere - ionosphere coupling is unaffected, the solar wind - ionosphere coupling is affected of enhanced spatial resolution.

### 4.3   Local dynamics

Figures 4 and 5 show the time series of the IMF magnitude $|B|$ in the Geotail and Cluster orbits during the 2012 and 2014 events compared with the GUMICS-4 results along the satellite tracks. The standard deviations are computed using the same

methods as in section 4.1, and are given in panels a and b. Since the inner boundary of the GUMICS-4 MHD region is at 3.7 $R_E$, the times when Cluster is closer than 3.7 $R_E$ to Earth are ignored when computing SD values.

Prior to the arrival of the sheath region in 2012, Geotail enters the plasma sheet boundary layer earlier than predicted by GUMICS-4. During the ICME sheath there are many dips and peaks in both plots, with the difference between measured (both Geotail and Cluster) and predicted values varying, as can be seen from figures 4a and 4b. Also, Figure 4a shows that

starting from July 17, 06:00 UT the measured field at Geotail increases as the satellite goes to the magnetosheath proper, while GUMICS-4 prediction decreases as the orbit track in GUMICS-4 approaches the shock region (see Figure 3a). The 2014 event shows similar features especially when Geotail enters and exits the magnetosphere at 23:14 UT, April 28, and at 12:00 UT, April 30, respectively, with measured (by Geotail) $|B|$ in the former case fluctuating and rising sharply from 10 nT to 40 nT while the GUMICS-4 $|B|$ increases more steadily from a few nT to 20 nT as the satellite enters from the magnetosheath to

the magnetosphere. In the latter case decrease (increase) of measured (simulated) $|B|$ occurs several hours after the spacecraft exits the magnetosphere (later yellow-shaded region in Figure 5a) because of the differences in the moment of exit (and exact location of the magnetopause location). Note that while Cluster makes an entry into the magnetosphere at 16:12 UT, April 29, GUMICS-4 predicts a position within the magnetosheath and an entry into the magnetosphere only following the end of the

cloud.

Note that the Cluster perigee (2 $R_E$) (Figure 4c) is below the inner boundary of the GUMICS-4 simulation (3.7 $R_E$), which causes the simulation field to record unphysical values around the time of the maxima at 09:00 on July 14, 2012 and 15:00 on July 16, 2012, hence the data gaps in GUMICS-4 data plots.

The effect of the ICME sheath is visible after its arrival in Figure 4, with both measured and predicted $|B|$ fluctuating. The

ICME magnetic cloud proper seems to cause largest difference in $|B|$ during the 2012 event, when the driving was quite strong. The standard deviations (SD) over the simulated time ranges using 0.5 $R_E$ spatial resolutions are considerably lower on Geotail orbit (2012: 5.476, 2014: 6.564) than on Cluster orbit (2012: 25.054, 2014: 24.795).

## 5   Discussion

In this paper we study 1) how the magnetosphere responds to two ICME events with different characteristics by means of

using the GUMICS-4 global MHD simulation, and 2) how accurately GUMICS-4 reproduces the effects of the two events. The 2012 event was stronger in terms of solar wind driver, the 2014 event being significantly weaker both in terms of solar wind speed and IMF magnitude. We considered both global and local parameters, including magnetopause nose position along the Sun-Earth line, total energy transferred from the solar wind into the magnetosphere, and the ionospheric cross-polar cap potential (CPCP). Local measures include response of the magnetic field magnitude along the orbits of Cluster and Geotail

spacecraft. The two ICME events were simulated using 0.5 $R_E$ maximum spatial resolution. To test the effect of grid resolution enhancement on global dynamics, we simulated 6-hour subsets of both CME cloud periods with 0.25 $R_E$ maximum spatial resolution. As an uncertainty metrics we use both relative difference magnitude $\delta$ and standard deviation SD.

Due to stronger solar wind driving, the 2012 event causes the magnetosphere to compress more than during the 2014 event, with the magnetopause moving Earthward at the onset of the 2012 ICME sheath and reaching 7 $R_E$ distance from Earth, until

moving Sunward at the onset of ICME magnetic cloud (see figure 6c). Both ICMEs are preceded by low IMF $B_Z$ and solar wind dynamic pressure, with the 2014 missing high amplitude fluctuations before ICME cloud due to absence of separate ICME sheath. Despite this, the movement of the magnetopause is similarly Earthward prior to the cloud, reaching 9.5 $R_E$ just before the onset of the cloud (see figure 7c). During the cloud however, the orientation of the IMF slowly rotates from southward to northward and the magnetopause is in constant Sunward (Earthward) motion in 2012 (2014). While the polarity

of the IMF changes before the end of the ICME in 2012, it changes from southward to northward only after the end of the ICME in 2014.

The magnetopause nose location in GUMICS-4 is identified as a single grid point from the maximum value of $J_Y$ along the Sun-Earth line. Location deviations in response to solar wind driving in the GUMICS-4 results is dependent on the driver

intensity: Stronger driving during the 2012 CME magnetic cloud leads to larger relative difference magnitude $\delta$ (2012: 8.0% $\delta$ on average) as compared to the Shue et al. (1997) model, whereas the agreement between the simulation and the empirical model is quite good (3.3% $\delta$ on average) during weaker driving during the 2014 event (figures 6 and 7). This view is further supported by standard deviations (SD): For the full simulation time range SD is 0.661 (0.321) in 2012 (2014). Average $\delta$ during nominal solar wind conditions is almost identical for both events: 2.5% for the 2012 event and 2.4% for the 2014 event.

Comparison of the magnetopause location between the 0.25 $R_E$ (0.5 $R_E$) resolution run and the Shue model show that the relative difference between the two is below 10% 92% (89%) of the 6 hour subset in 2012 (Figure 6c), while corresponding analysis of the 6 hour subset in 2014 (Figure 7c) yielded differences below 10% 100% of the time regardless of the resolution. It should be noted that, despite the relative difference magnitude is slightly lower for the 0.5 $R_E$ resolution run than for the 0.25 $R_E$ resolution run for both the 2012 (4.9% and 5.6%) and the 2014 (3.2% and 4.5%) events, the 0.25 $R_E$ run reaches better agreement with the Shue model especially when the magnetopause is moving during high solar wind driving in July 16, 01:00 UT (Figure 6c).

When spatial resolution is increased, gradient quantities such as $J_Y$ have sharper profiles and therefore larger values (Janhunen et al., 2012). As it is the maximum value of $J_Y$ that we use to locate the magnetopause nose, the nose position evaluation in the lower resolution runs is more ambiguous both due to the larger spread of the current and due to the larger grid cell size. This may lead to changes in the maximum value up to several $R_E$ over short time periods in response to upstream fluctuations. In the finer resolution runs, $J_Y$ distribution is sharper, which leads to lesser fluctuations in the maximum value determination. However, the differences between the two grid resolutions occur only under rapidly varying solar wind or very low solar wind density conditions.

The empirical models developed by Shue et al. (Shue et al., 1997, 1998) are based on statistical analysis of large number of spacecraft measurements of plasma and magnetic field during magnetopause crossings. While the Shue et al. (1997) model is optimized for moderate upstream conditions, the Shue et al. (1998) targets especially stronger driving periods. However, we computed the difference in the magnetopause position between the two models and found that it is mostly less than 0.1 $R_E$ with maximum difference of 0.4 $R_E$, with Shue et al. (1997) model predicting more sunward magnetopause nose. Because of the small difference at the magnetopause nose, we have only used Shue et al. (1997) model in our study. Our results agree with previous papers (Palmroth et al., 2003; Lakka et al., 2017), with the latter reporting 3.4% average relative difference between the Shue model and GUMICS-4. Moreover, according to Gordeev et al. (2015), global MHD models are very close to each other in terms of predicting magnetopause standoff distance.

Differences in the magnetopause location do not necessarily translate into differences in global measures, as can be seen from figures 6d and 7d, which show the time evolution of the energy transferred from the solar wind through the magnetopause surface. The response of the total energy $E_{tot}$ during both ICME cloud periods is quite similar regardless of the used grid resolution. As an integrated quantity, energy entry is a better indicator of the true physical processes of GUMICS-4 solution and does not suffer from dependence on grid resolution like the maximum $J_Y$ (Janhunen et al., 2012). Therefore, in analyses of simulation results, it would be better to consider such global integrated quantities, even if they have no direct observational counterparts. This can be seen in figures 6d and 7d, with large differences between GUMICS-4 and the $\epsilon$-parameter (Perreault

and Akasofu, 1978) in energy transferred from the solar wind into the magnetosphere in both 2012 and 2014. However, standard deviations show that GUMICS-4 reproduces temporal evolution of the $\epsilon$-parameter better during low solar wind driving (2014) than during high driving (2012), as the respective SD values are 0.725 and 2.263. Moreover, our results are mostly of the same order of magnitude compared to what was obtained by Palmroth et al. (2003) by using plasma flow streamlines for computing the magnetopause surface from GUMICS-4 results.

In the ionosphere, the cross-polar cap potential value is dependent on the grid resolution, with higher resolution yielding higher polar cap potential values (see Figures 6e and 7e). In comparison with the PCI index (Ridley and Kihn, 2004), standard deviation is considerably lower for the 2014 event (5.107) than for the 2012 event (15.838). Thus, at least two factors contribute to the ionospheric coupling: Grid resolution and intensity of solar wind driving. Considering that the SD values are clearly higher than e.g. the corresponding energy transfer values, and that the PCI index considers only the northern hemisphere, the PCI index may not provide the most accurate reference for GUMICS-4. However, both considerable difference between GUMICS-4 and the PCI and the dependence on grid resolution agree with previous studies (e.g. Lakka et al., 2018). Generally, global MHD codes differ from each other in terms of the CPCP values (Gordeev et al., 2015). It is not easy to reproduce realistic CPCP values in a global MHD code, since they are generally prone to close excessive amount of electric current through the polar cap and thus the CPCP values are either unrealistically large (e.g. LFM model (Lyon et al., 2004)), with reasonable auroral electrojet currents, or reasonable accompanied by low auroral electrojet currents (De Zeeuw et al., 2004) (e.g. GUMICS-4 and BATS-R-US model (Powell et al., 1999)).

The polar cap structure and the distribution of the FAC do not change much in either of the simulations, thus suggesting that the coupling of the magnetosphere and the ionosphere remains relatively constant. As is shown in figures 10a–10b, the region 1 currents are clearly visible, while the region 2 currents get stronger only by enhancing the grid resolution in the MHD region (Janhunen et al., 2012). However, the upstream conditions change considerably from 01:00 to 03:00, with the upstream Alfvén Mach number decreasing from 1.9 to 0.6, suggesting that polar cap potential saturation mechanisms are likely to take place (Ridley, 2007; Wilder et al., 2015; Lakka et al., 2018). Considering that GUMICS-4 reproduces saturation with both 0.5 $R_E$ (this paper) and 0.25 $R_E$ resolutions (Lakka et al., 2018), it is apparent that the FAC influence on the dayside magnetospheric magnetic field do not contribute to the saturation effect. However, to actually prove it is beyond te scope of the current paper. We therefore conclude that the increase of the CPCP during the 0.5 $R_E$ simulation run is caused by processes outside of the magnetosphere, likely in the magnetosheath, and that GUMICS-4 responds differently to low Alfvén Mach number solar wind depending on grid resolution.

Figures 8 and 9 illustrate the CPCP as a function of the solar wind $E_Y$ component. Color-coded are the IMF magnitude in figures 8a and 9a, the solar wind speed in figures 8b and 9b, and the upstream Alfvén Mach number in figures 8c and 9c. Nominal solar wind conditions before and after the actual ICME events as well as the ICME sheath and cloud periods are considered separately. We note that only results from the lower spatial resolution (0.5 $R_E$) runs are included in the figures. Consistent with earlier studies, Figure 8 shows saturation of the CPCP during high solar wind driving (see e.g. Shepherd (2007); Russell et al. (2001)): With nominal solar wind conditions or during ICME sheath period the response of the CPCP to the upstream $E_Y$ is rather linear, while for ICME cloud period the CPCP saturates, when $E_Y > 5\mathrm{mV/m}$. From Figure 8a it

can be seen that the saturation occurs when $B > 12$ nT and Figure 8b shows that the increase of the CPCP in the linear regime depends on the upstream velocity in such a way that the increase is clearly higher for lower velocity values (cloud event), than for higher velocity values (sheath event and nominal conditions), as suggested by previous statistical (Newell et al., 2008) and numerical (Lopez et al., 2010) studies. The latter study proposes that this is because of the more rapid diversion of the solar

wind flow in the pressure gradient dominated magnetosheath under faster solar wind, which leaves a smaller amount of plasma at the magnetic reconnection site.

The saturation of the CPCP is absent in Figure 9 due to the significantly weaker solar wind driving during the 2014 event (the upstream $E_Y$ is below 4 mV/m). This in turn leads to the upstream Alfvén Mach number to be on average 5.8 during the ICME cloud event. Lavraud and Borovsky (2008) suggests that when the Alfvén Mach number decreases below 4 and

the overall magnetosheath plasma beta ($p/p_B$, where $p$ is the plasma pressure and $p_B$ the magnetic pressure) below 1, the magnetosheath force balance changes such that plasma flow streamlines are diverted away from the magnetic reconnection merging region in the dayside magnetopause (Lopez et al., 2010), which causes the CPCP saturation. However, the CPCP saturation limit of $M_A = 4$ is not necessarily the only governing parameter, as there is both observational evidence with large $M_A$ values (up to 7.3) (Myllys et al., 2016) and simulation results indicating saturation at low but above $M_A = 1$ values (this

study). Nonetheless, our results suggest that the saturation of the CPCP is dependent on the upstream $M_A$ in such a way that $M_A$ needs to be below 4 for the saturation to occur.

An interesting aspect is that the CPCP does not reach its maximum simultaneously with $E_Y$, i.e. the CPCP is largest with moderate $E_Y$ (5–6 mV/m) (see Figure 8). As $E_Y$ increases to 11 mV/m, the CPCP decreases from 70 kV to 40 kV. This is actually apparent in Figure 1h as well: The absolute values of both $B_Z$ and $V_X$ reach their maximum values a few hours

after the onset of the magnetic cloud, which is at 6.54 UT, July 15. However, the CPCP is at that time quite moderate, about 40 kV, and does not reach its maximum until July 16, when both $B_Z$ and $V_X$ have already reduced significantly. Thus the CPCP overshoots in Figure 8, a feature that was not observed in a GUMICS-4 study by Lakka et al. (2018) using artificial solar wind input consisting of relatively high density and constant driving parameters.

The performance of GUMICS-4 was put to test by means of comparing the magnetic field magnitude $|B|$ to in-situ data of

Cluster and Geotail satellites. GUMICS-4 values are mostly lower than those measured by either of the two spacecraft, with GUMICS-4 predictions being closer to Cluster than Geotail. Computed standard deviations reveal that, over the entire simulation periods, the temporal evolution of GUMICS-4 magnetic field magnitude predictions is closer to Geotail measurements (2012: SD = 5.476, 2014:SD = 6.564, equatorial orbit) than Cluster measurements (2012: SD = 25.054, 2014: SD = 24.795, polar orbit) for both events. It should be noted that the times when Cluster is closer than 3.7 $R_E$ to Earth are ignored when

computing SD values due to the inner boundary of the GUMICS-4 MHD region, which is located at 3.7 $R_E$.

During both events, $|B|$ is increased during ICMEs, especially their magnetic cloud counterparts. During the 2012 ICME sheath both Cluster and Geotail record fluctuating $|B|$ until the onset of the cloud. Albeit missing sheath in 2014, magnetic field magnitude measured by Cluster fluctuates as well prior to the cloud. At the same time (April 29, 15:00 UT) $|B|$ measured by Geotail decreases sharply. The difference between Cluster/Geotail and GUMICS-4 is mostly order of 10%, but can

reach above 50% values especially during the 2012 magnetic cloud event in both Cluster and Geotail orbit. Such difference

seem relatively large especially since it was shown by Ridley et al. (2016) that all the global MHD models available at the Community Coordinated Modeling Center (CCMC) are close to each other when comparing the ability to reproduce magnetic field components to in-situ measurements. Albeit the study used 662 simulation runs, it should be noted that GUMICS-4 was used in only 12 of them. However, GUMICS-4 should predict $|B|$ closer to in-situ measurements at least during moderate solar wind driving, as was shown by Facskó et al. (2016). In his work the difference in |B| was 10% or lower on February 20 2002, when no ICME events were recorded.

With such discrepancy between our results and previous results, we checked some of the simulation runs at CCMC, in which BATS-R-US (Powell et al., 1999) code was used, and searched for runs of either of the two ICME events discussed in this paper, with magnetic field measurements along Geotail and/or Cluster orbit also available. BATS-R-US was chosen since it shares several features wich GUMICS-4. We found one simulation run (CCMC run name Tom_ Bridgeman_ 022415_ 1) in which the 2012 event was simulated, with results along Geotail orbit available. In addition, we simulated the 2014 event (CCMC run name Antti_ Lakka_ 070918_ 2) to check the results along Cluster path. Consequently, we are able to compare GUMICS-4 and BATS-R-US in both 2012 (Geotail) and 2014 (Cluster), and the results are shown in figure 11. Panel a shows comparison between the two models during the 2012 event, and panel b during the 2014 event. In-situ measurements by Geotail (Cluster) are shown in panel a (b). Note that the 2012 BATS-R-US run was completed at around July 17 00:00 UT. By looking at the figure it is apparent that the predictions of both GUMICS-4 and BATS-R-US are quite similar especially during the magnetic cloud events at both Cluster and Geotail orbits. Actually, GUMICS-4 is mostly closer to Cluster measurements than BATS-R-US in 2014, when Cluster exits the magnetosphere and $|B|$ measured by Cluster fluctuates between 10 nT to 40 nT, as was discussed in section 4.3. In 2012 large difference in $|B|$ (up to 100%) during ICME cloud applies to both models. During ICME sheath and nominal solar wind conditions $|B|$ fluctuates more and the prediction accuracy of the models depends on the time interval under inspection. It is evident that both models are quite equal considering the ability to reproduce $|B|$ during both 2012 and 2014 ICME events.

The discrepancy between in-situ measurements and the two models may not concern only GMHD models, since we computed the magnetic field during the 2012 event at Cluster orbit using the empirical Tsyganenko magnetic field model T89 (e.g. Tsyganenko and Sitnov, 2005). For most parts GUMICS-4 is actually closer to Cluster observations, with the gap between the two models gradually decreasing as Cluster approaches the perigeum on July 16 (not shown). Therefore it is reasonable to assume that something in the ICME event, possibly unusually strong compression, leads to larger field than predicted by the GMHD models or the Tsyganenko model, and that e.g. increasing spatial resolution of the GMHD models would not make significant difference for the two reasonably similar codes (Janhunen et al., 2012). The negligible effect of enhanced spatial resolution is actually shown in figures 4 and 5 for GUMICS-4.

It should be noted that the event is one of the strongest that occurred in 2012 by the mean magnetic field magnitude value during magnetic cloud. On the other hand, in some cases good agreement can be obtained when modelling strong ICMEs. Recently Kubyshkina et al. (2019) studied two events that occurred in 2015 and were the strongest events of solar cycle 24, and achieved reasonable agreement between measurements and different models, such as BATS-R-US with Rice Convection Model, and empirical models including Tsyganenko T96 model. Mostly the error in the magnetic field magnitude was less than

15 nT, with the error increasing for short while to more than 50 nT. The reason why some events cause greater errors than other events is however out of the scope of the current paper and is left for future studies.

We conclude that for both events, $|B|$ predicted by GUMICS-4 is closer to Cluster observations, which feature high magnetic field magnitude outside the plasma sheet. While the differences between GUMICS-4 and in-situ measurements can be quite

large, it was shown that the $|B|$ predicted by GUMICS-4 agrees well with BATS-R-US predictions, and thus the large differences are not model-related, but rather related to the upstream conditions during the ICME events. Thus the relative difference in $|B|$ may not be good metrics when simulating ICME events and evaluating the performance of a global MHD model.

While the agreement between predicted and measured $|B|$ may depend on the usptream conditions, the overall time evolutions seem to have a better match, and the SD values suggest that GUMICS-4 reproduces temporal evolution of $|B|$ better at Geotail

orbit, which is much further away from the Earth than Cluster, and resides mostly in the lobe and on the boundary layer. We computed standard deviations for Cluster orbit when the S/C is both further and closer than 5 $R_E$ away from the center of the Earth. SD for further than 5 $R_E$ is 22.984 (19.666) for the 2012 (2014) event, while for closer than 5 $R_E$ the SD is 106.337 (104.605) for the 2012 (2014) event. If these calculations are repeated for 6 $R_E$ distance, the SD values are 14.390 (15.282) when the S/C is further in 2012 (2014), and 104.618 (88.423) when the S/C is closer in 2012 (2014). Thus, the temporal

evolutions agree better when Cluster is further away from the Earth.

The differences are most likely not caused by grid cell size variations due to the adaptive grid of GUMICS-4, because the simulation runs over simulated 6-hour stages produce quite similar results for both resolutions. Also, the two runs deviates most from each other during the first hours of the 6 hour stage, during which the 0.25 $R_E$ run may not have fully eliminated the effects of simulation initialization, which can prevail hours (Lakka et al., 2017). Moreover, the adaptive grid of GUMICS-4 is

enhanced the most near the dayside magnetopause. Both events show signs of increased deviation from the measurements near the dayside magnetopause (edges of yellow-shaded regions in figures 4 and 5), further manifesting inaccuracies in determining the magnetopause in GUMICS-4.

## 6 Conclusions

The results of this paper can be summarized as follows:

(1) Enhancing spatial resolution of the magnetosphere in GUMICS-4 affects the accuracy of the determination of the the magnetopause subsolar point. Global measures, such as energy transferred from the solar wind into the magnetosphere, are not affected. The cross-polar cap potential can be affected significantly, with up to over factor of 2 difference between simulations using different spatial resolutions for the magnetosphere.

(2) Our results show signs of cross-polar cap potential saturation during low upstream Alfvén Mach number. GUMICS-4

responds differently to low Alfvén Mach number solar wind, which may affect the saturation phenomena. This may lead to grid size effects to polar cap saturation in MHD simulations.

(3) Comparison metrics choice should be done cautiously. For instance, relative difference in $|B|$ may not be a good metrics

when studying ICME events. Due to inaccuracies in the magnetopause subsolar point determination, comparison between GUMICS-4 and in-situ data should be done cautiously when the spacecraft is near the magnetopause.

*Data availability.* Solar wind data are freely available from the NASA/GSFC Omniweb server (https://omniweb.gsfc.nasa.gov/). Solar energetic particle data are freely available from the NOAA NCEI Space Weather data access (https://www.ngdc.noaa.gov/stp/satellite/goes/index.html).

5 *Competing interests.* The authors declare that they have no conflict of interest.

*Acknowledgements.* The calculations presented above were performed using computer resources within the Aalto University School of Science "Science-IT" project. This project was funded by the Academy of Finland grants #1267087, #288472, and #310444. We acknowledge use of NASA/GSFC's Space Physics Data Facility's OMNIWeb service, and OMNI data. Solar energetic particle data supplied courtesy of ngdc.noaa.gov.

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

**Table 1.** Summary of the event simulations within the current study.

| Event year | Nominal solar wind [h] | Event date and time | Event length [h] | Resolution [$R_E$] |
|---|---|---|---|---|
| 2012 | 9.9 | 18:53 UT, July 14 – 04:19 UT, July 17 | 57.4 | 0.5 |
| 2014 | 25.6 | 20:38 UT, April 29 – 17:51 UT, April 30 | 21.2 | 0.5 |
| 2012 | 0 | 21:00 UT, July 15 – 03:00 UT, July 16 | 6 | 0.25 |
| 2014 | 0 | 00:00 UT, April 30 – 06:00 UT, April 30 | 6 | 0.25 |

**Table 2.** Average relative difference magnitudes in the magnetopause nose position for given simulation phase.

| Event year | Resolution [$R_E$] | Nominal SW [%] | Sheath [%] | Cloud [%] | 6 hours [%] |
|---|---|---|---|---|---|
| 2012 | 0.5 | 2.5 | 4.5 | 8.0 | 4.9 |
| 2014 | 0.5 | 2.4 | - | 3.3 | 3.2 |
| 2012 | 0.25 | - | - | - | 5.6 |
| 2014 | 0.25 | - | - | - | 4.5 |

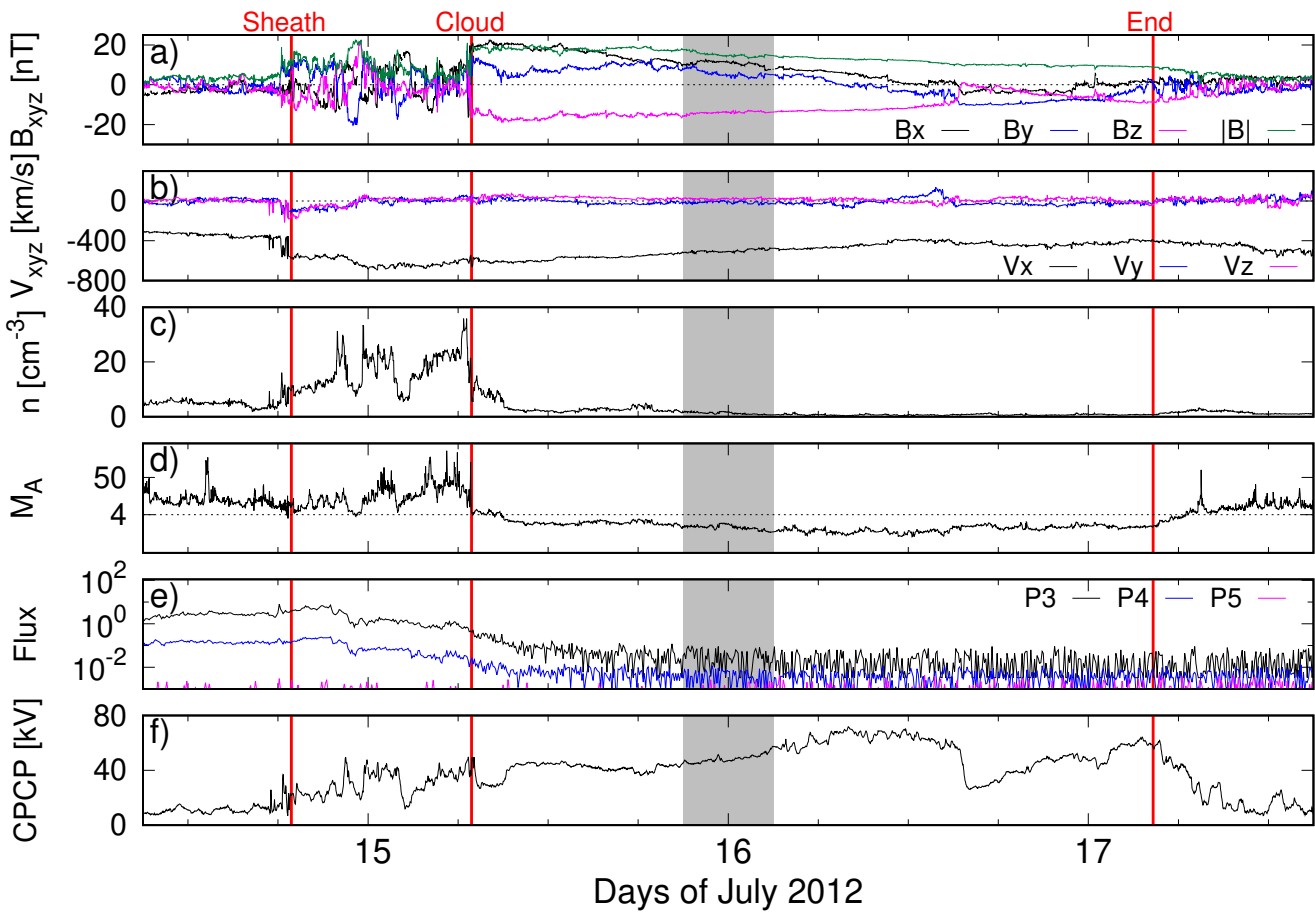

**Figure 1.** Solar wind and IMF conditions during July 14 09:00 UT – July 17 15:00 UT, 2012. Panels from top to bottom: a) IMF components $B_X$, $B_Y$ and $B_Z$ and the IMF magntiude in nT, b) plasma velocity components $V_X$, $V_Y$ and $V_Z$ in km/s, c) plasma number density $n$ in cm$^{-3}$, d) upstream Alfvén Mach number $M_A$ ($M_A = 4$ is marked with dotted line), e) GOES-15 geostationary orbit proton fluxes for three energy channels between 8–80 MeV, and f) the ionospheric cross-polar cap potential from GUMICS-4. Data in panels a–d is measured by ACE/Wind. Vertical red lines indicate onset of the ICME sheath/magnetic cloud or the end of the ICME event. Grey background shows the part of the ICME event that is simulated using both 0.25 and 0.5 R$_E$ as a maximum spatial resolution.

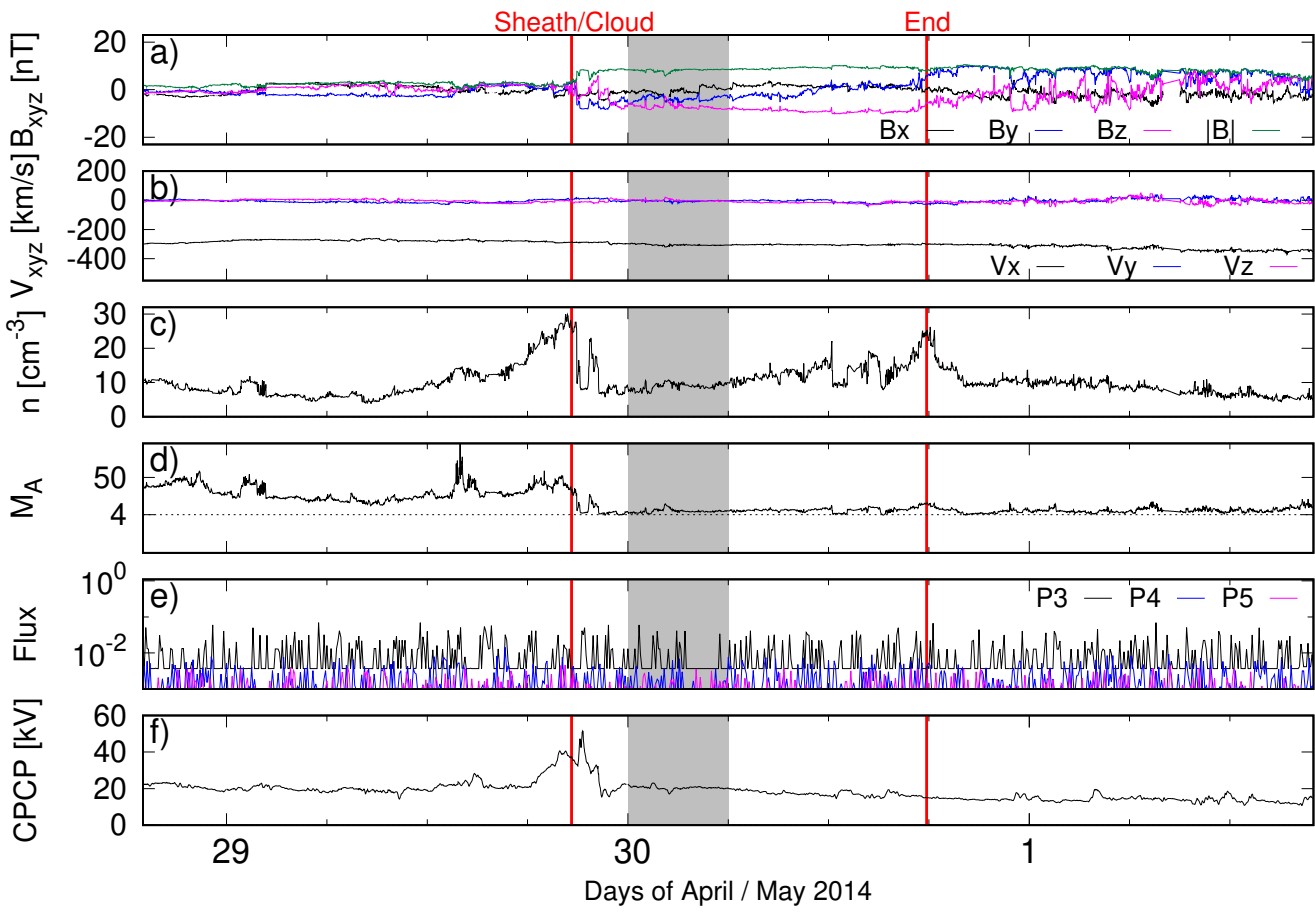

**Figure 2.** Solar wind and IMF conditions during April 28 19:00 UT – May 1 17:00 UT, 2014. Panels from top to bottom: a) IMF components $B_X$, $B_Y$ and $B_Z$ and the IMF magntiude in nT, b) plasma velocity components $V_X$, $V_Y$ and $V_Z$ in km/s, c) plasma number density $n$ in $cm^{-3}$, d) upstream Alfvén Mach number $M_A$ ($M_A = 4$ is marked with dotted line), e) GOES-15 geostationary orbit proton fluxes for three energy channels between 8–80 MeV, and f) the ionospheric cross-polar cap potential from GUMICS-4. Data in panels a–d is measured by ACE/Wind. Vertical red lines indicate onset of the ICME sheath/magnetic cloud or the end of the ICME event. Grey background shows the part of the ICME event that is simulated using both 0.25 and 0.5 $R_E$ as a maximum spatial resolution.

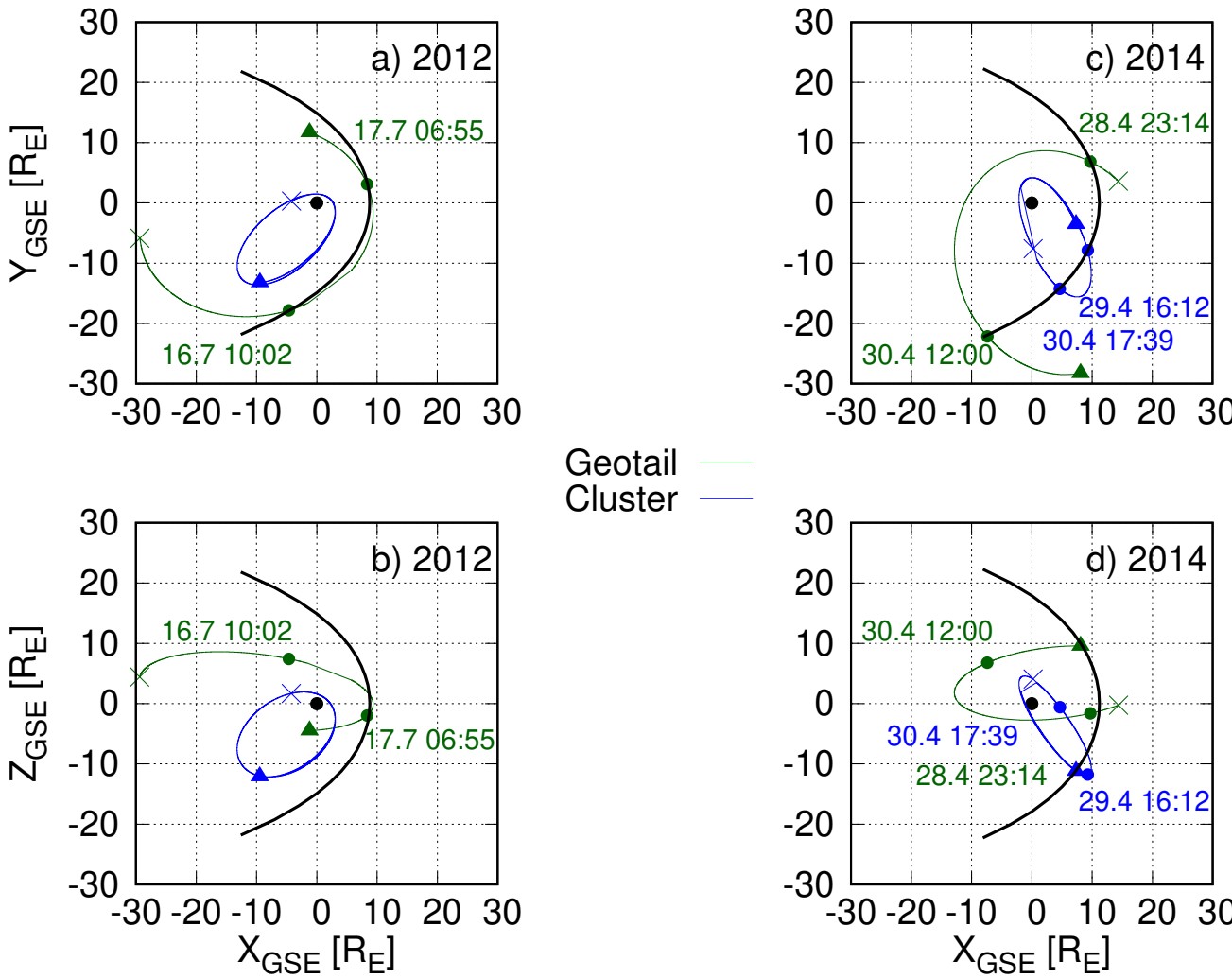

**Figure 3.** Orbits of Cluster 1 (blue) and Geotail (green) satellites during July 14 09:00 UT – July 17 15:00 UT, 2012 (panels a and b) and during April 28 19:00 UT – May 1 17:00 UT, 2014 (panels c and d). Orbits are shown on the $XY$ plane in panels a and c and on the $XZ$ plane in panels b and d. The coordinate system is GSE. The most earthward position of the Shue magnetopause during both time intervals is drawn in black. Start and end points of the time intervals are marked with a cross and a triangle, respectively. The points along the satellite orbits between which the spacecraft may encounter magnetopause crossings are marked with dots.

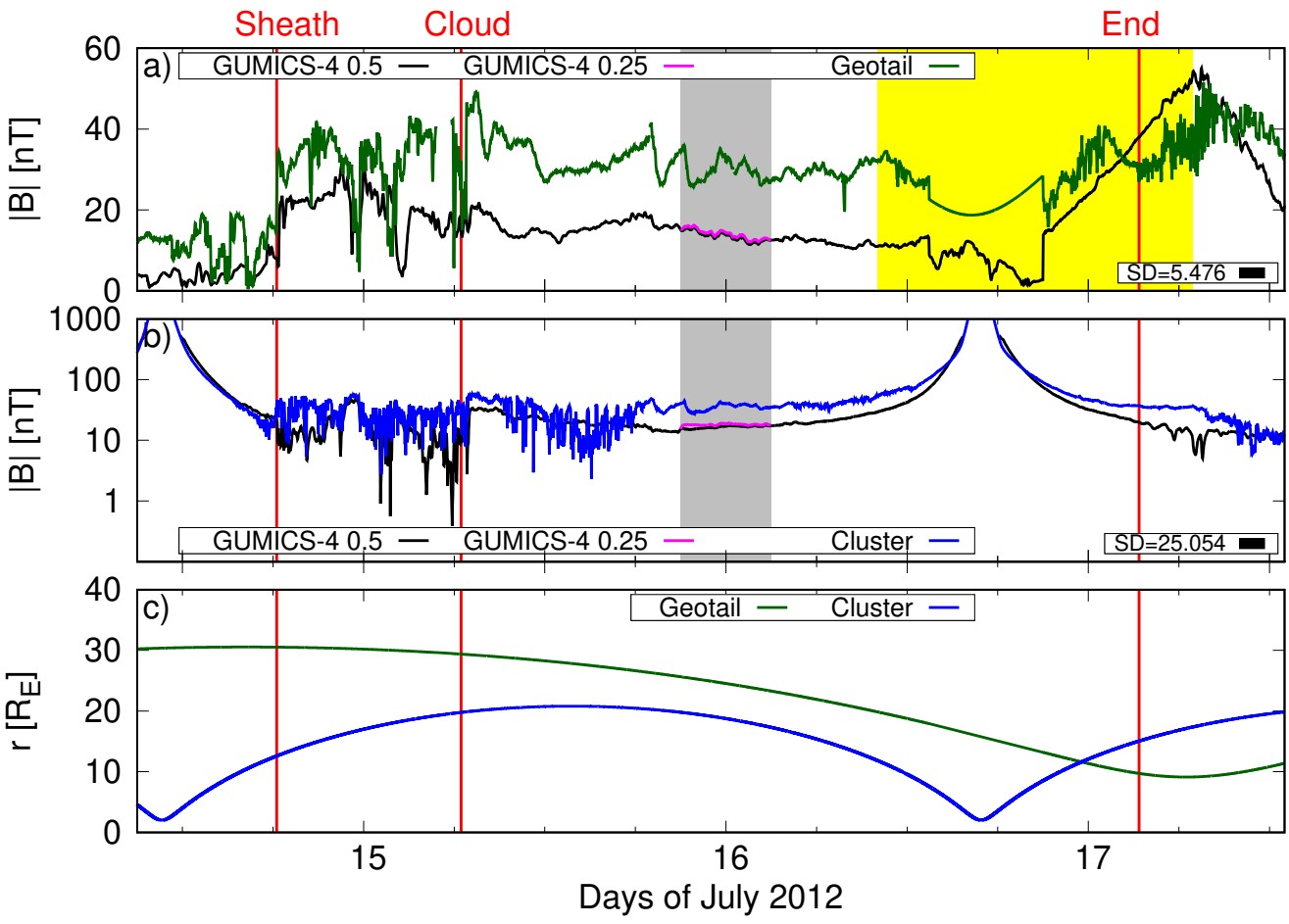

**Figure 4.** The time series of the magnetic field magnitude $|B|$ along the orbits of Geotail (panel a) and Cluster 1 (panel b) during July 14 09:00 UT – July 17 15:00 UT, 2012 as measured by Geotail (green) and Cluster 1 (blue) and predicted by GUMICS-4 (black and magenta). Black and magenta curves in panels a–b show GUMICS-4 results with maximum spatial resolution of 0.5 (black) and 0.25 (magenta) $R_E$. Panel c: Radial distance of both spacecraft from the center of the Earth. Yellow-shaded regions indicate approximate time intervals when satellite may exit the magnetosphere. Grey-shaded regions show the part of the ICME event simulated also using 0.25 $R_E$ maximum spatial resolution. Standard deviations (SD) for observation vs. GUMICS-4 (0.5 $R_E$ resolution) datasets are given in panels a and b.

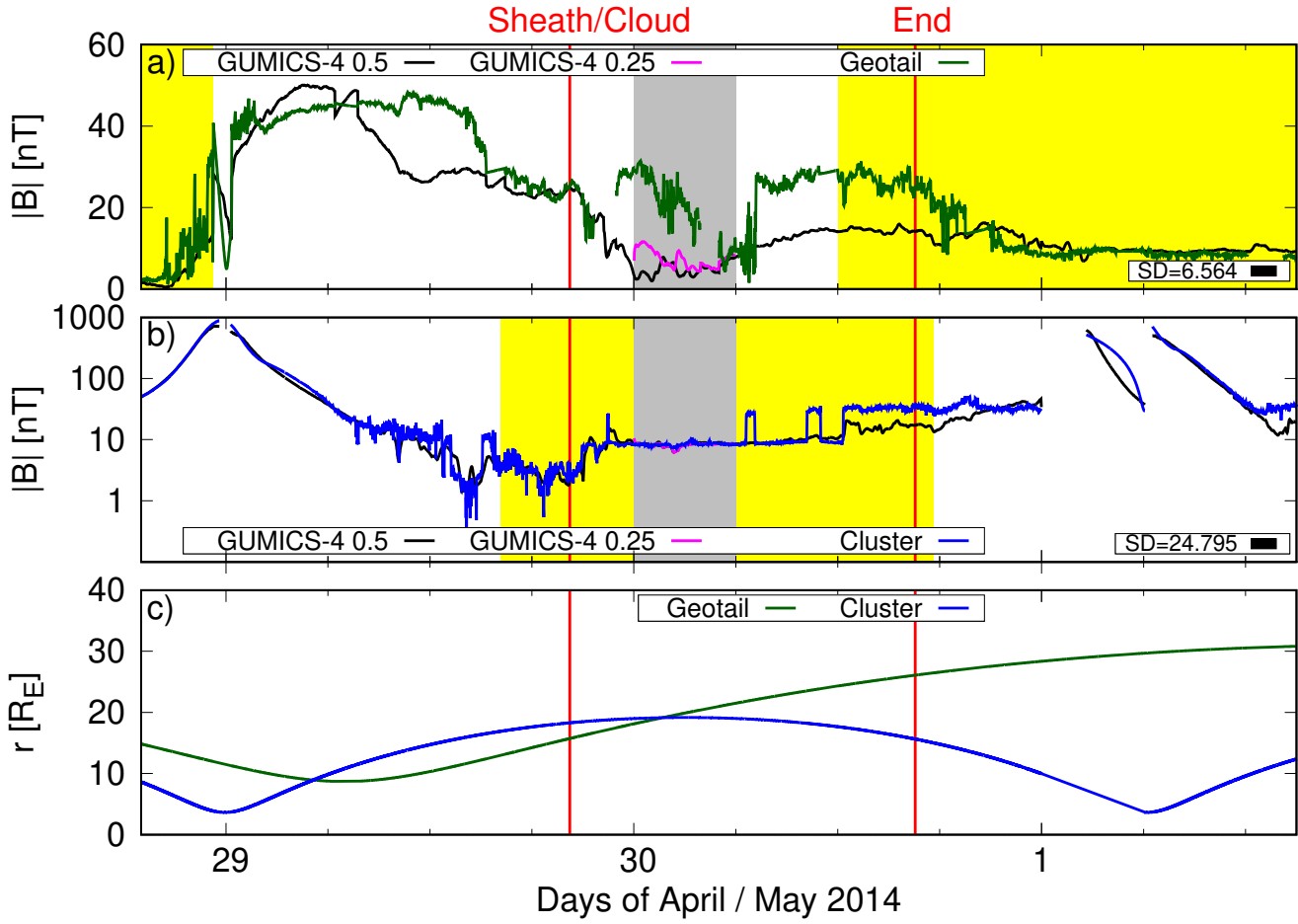

**Figure 5.** The time series of the magnetic field magnitude $|B|$ along the orbits of Geotail (panel a) and Cluster 1 (panel b) during April 28 19:00 UT – May 1 17:00 UT, 2014 as measured by Geotail (green) and Cluster 1 (blue) and predicted by GUMICS-4 (black and magenta). Black and magenta curves in panels a–b show GUMICS-4 results with maximum spatial resolution of 0.5 (black) and 0.25 (magenta) $R_E$. Panel c: Radial distance of both spacecraft from the center of the Earth. Yellow-shaded regions indicate approximate time intervals when satellite may exit the magnetosphere. Grey-shaded regions show the part of the ICME event simulated also using 0.25 $R_E$ maximum spatial resolution. Standard deviations (SD) for observation vs. GUMICS-4 (0.5 $R_E$ resolution) datasets are given in panels a and b.

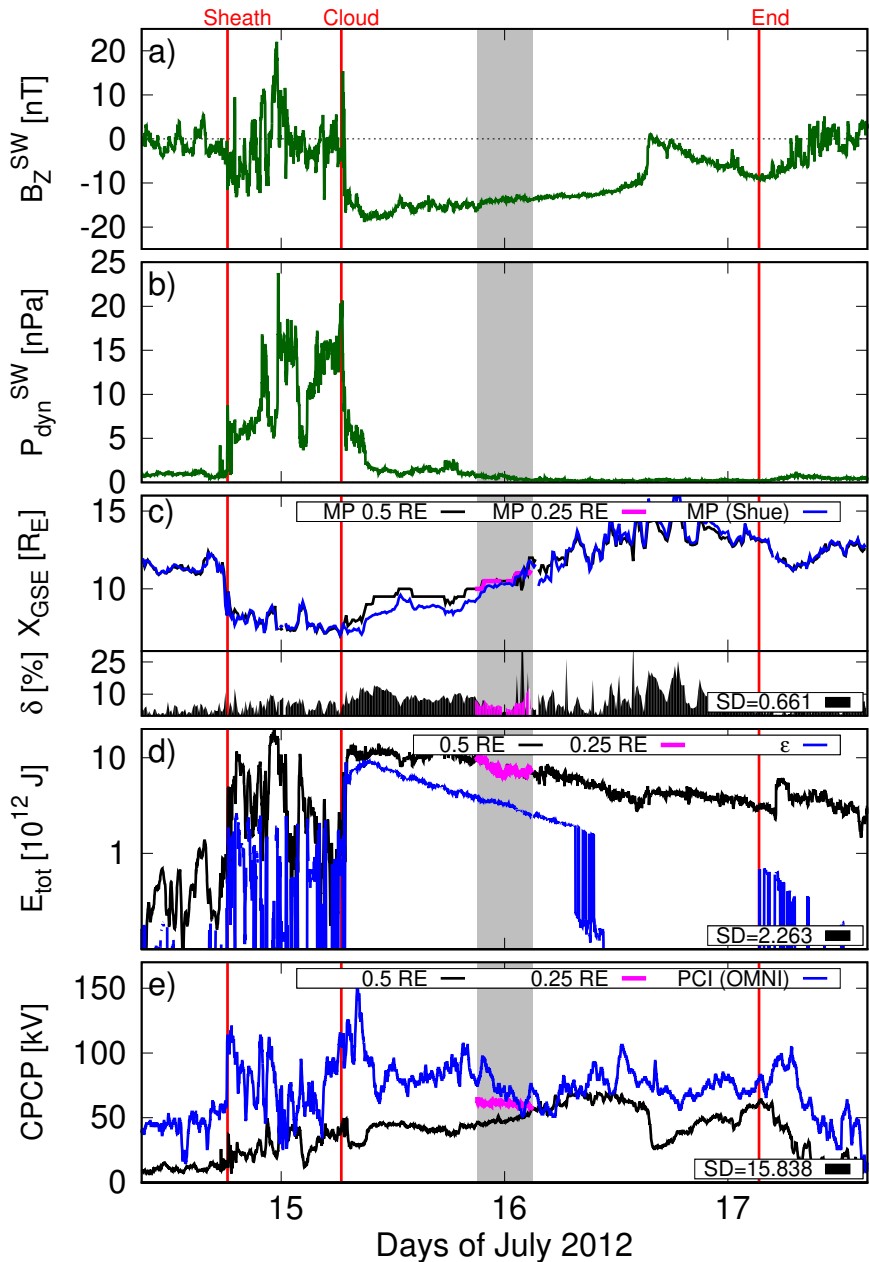

**Figure 6.** a) Interplanetary magnetic field $Z$-component, b) solar wind dynamic pressure, c) distance to the nose of the magnetopause, d) energy transferred from the solar wind into the magnetosphere through the dayside magnetopause, and e) the cross-polar cap potential during July 15 21:00 UT - July 16 03:00 UT, 2012. Magenta plots in panels c–d show results with maximum spatial resolution of 0.25 $R_E$. Blue curves in panels c, d, and e show the reference values (the Shue model, the $\epsilon$-parameter, the PCI index). The relative difference magnitude $\delta$ between GUMICS-4 and the reference value is shown in panel c. Standard deviations (SD) for reference vs. GUMICS-4 (0.5 $R_E$ resolution) datasets are given in panels c–e.

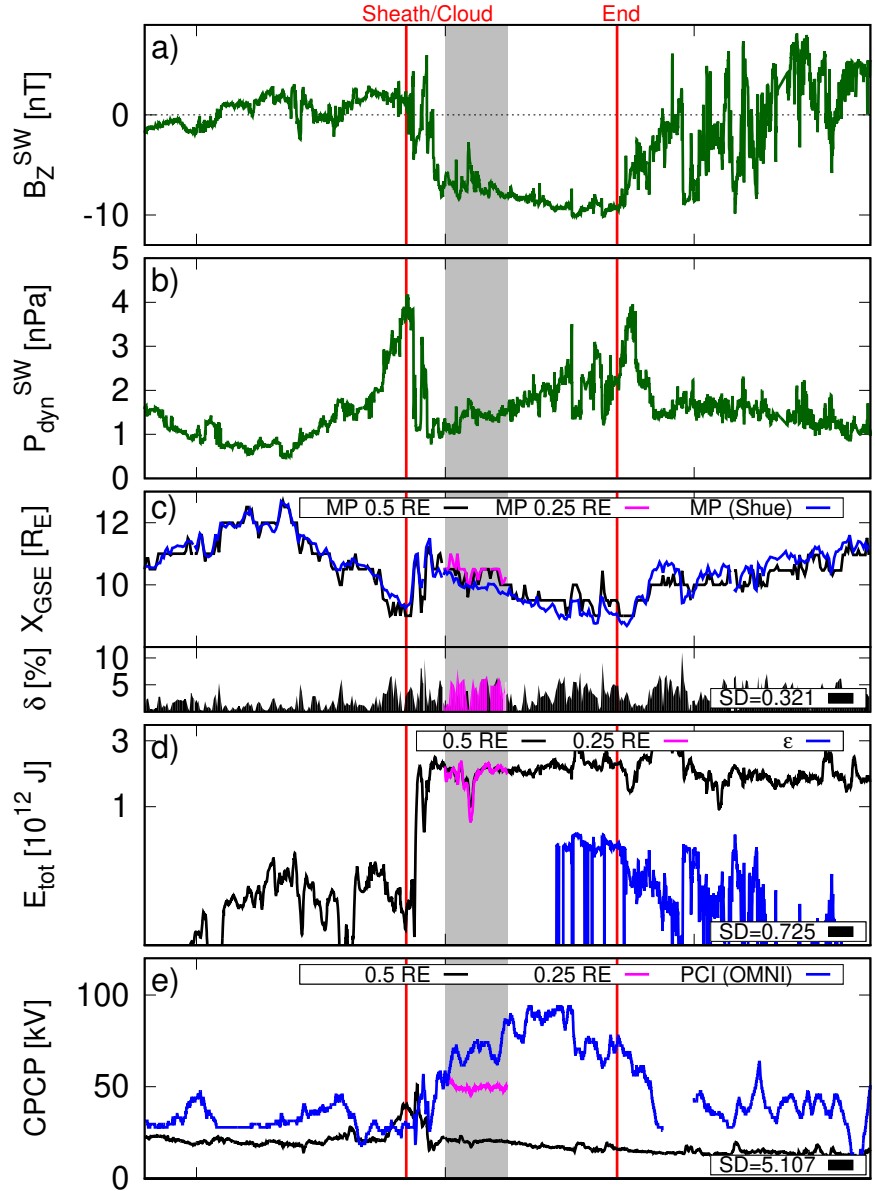

**Figure 7.** a) Interplanetary magnetic field $Z$-component, b) solar wind dynamic pressure, c) distance to the nose of the magnetopause, d) energy transferred from the solar wind into the magnetosphere through the dayside magnetopause, and e) the cross-polar cap potential during April 30 00:00 UT – 06:00 UT, 2014. Magenta plots in panels c–d show results with maximum spatial resolution of 0.25 $R_E$. Blue curves in panels c, d, and e show the reference values (the Shue model, the $\epsilon$-parameter, the PCI index). The relative difference magnitude $\delta$ between GUMICS-4 and the reference value is shown in panel c. Standard deviations (SD) for reference vs. GUMICS-4 (0.5 $R_E$ resolution) datasets are given in panels c–e.

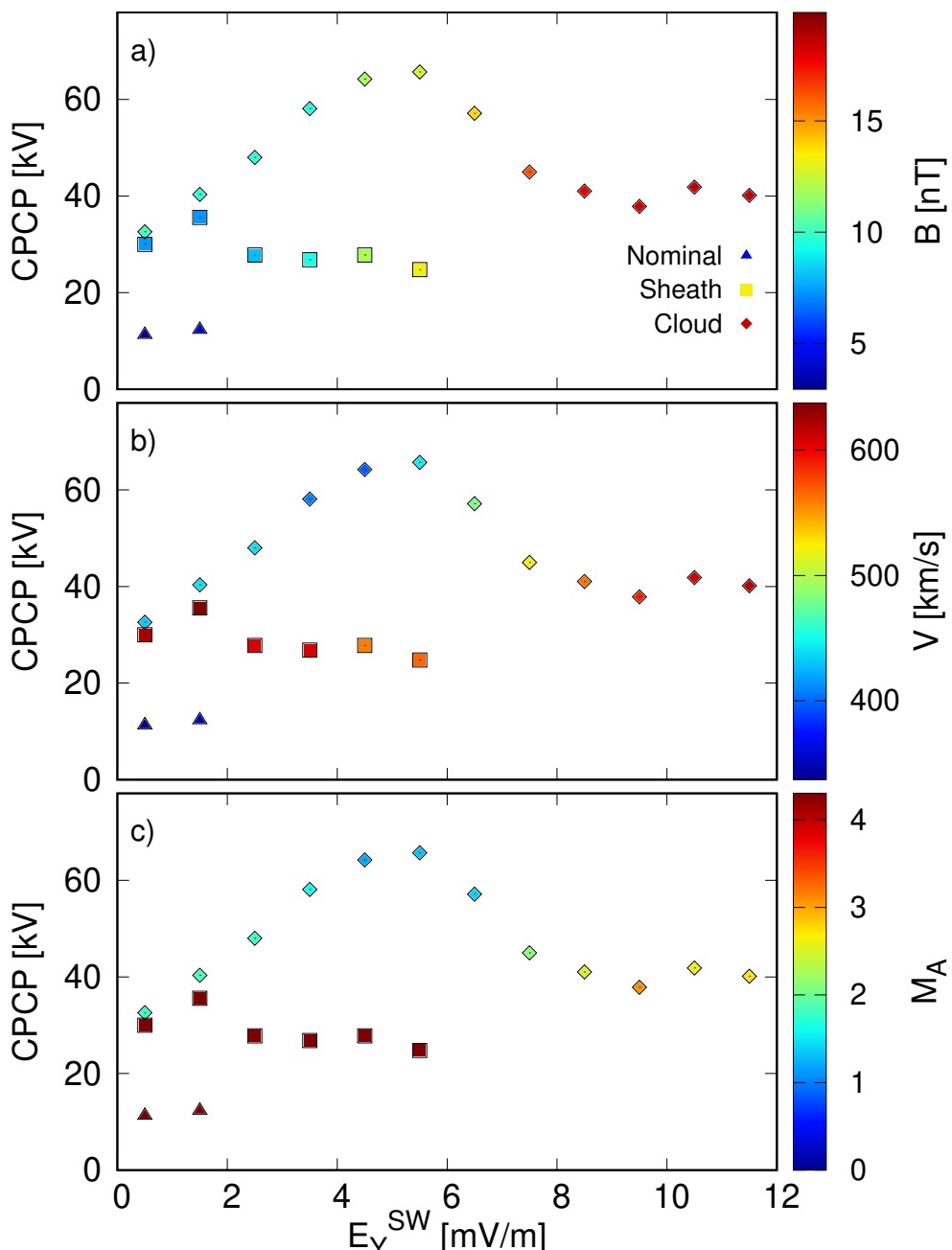

**Figure 8.** The cross-polar cap potential (CPCP) as a function of the IMF $E_Y$ for the 2012 ICME sheath and cloud periods, with nominal solar wind conditions before and after the ICME event taken into account separately. GUMICS-4 simulation data with 1 minute time resolution has been averaged by 10 minutes and binned by upstream $E_Y$ with $1.0 \, \mathrm{mV/m}$ intervals. Panels a, b and c show the magnitudes of the IMF, the upstream flow speed and the Alfvén Mach number, respectively.

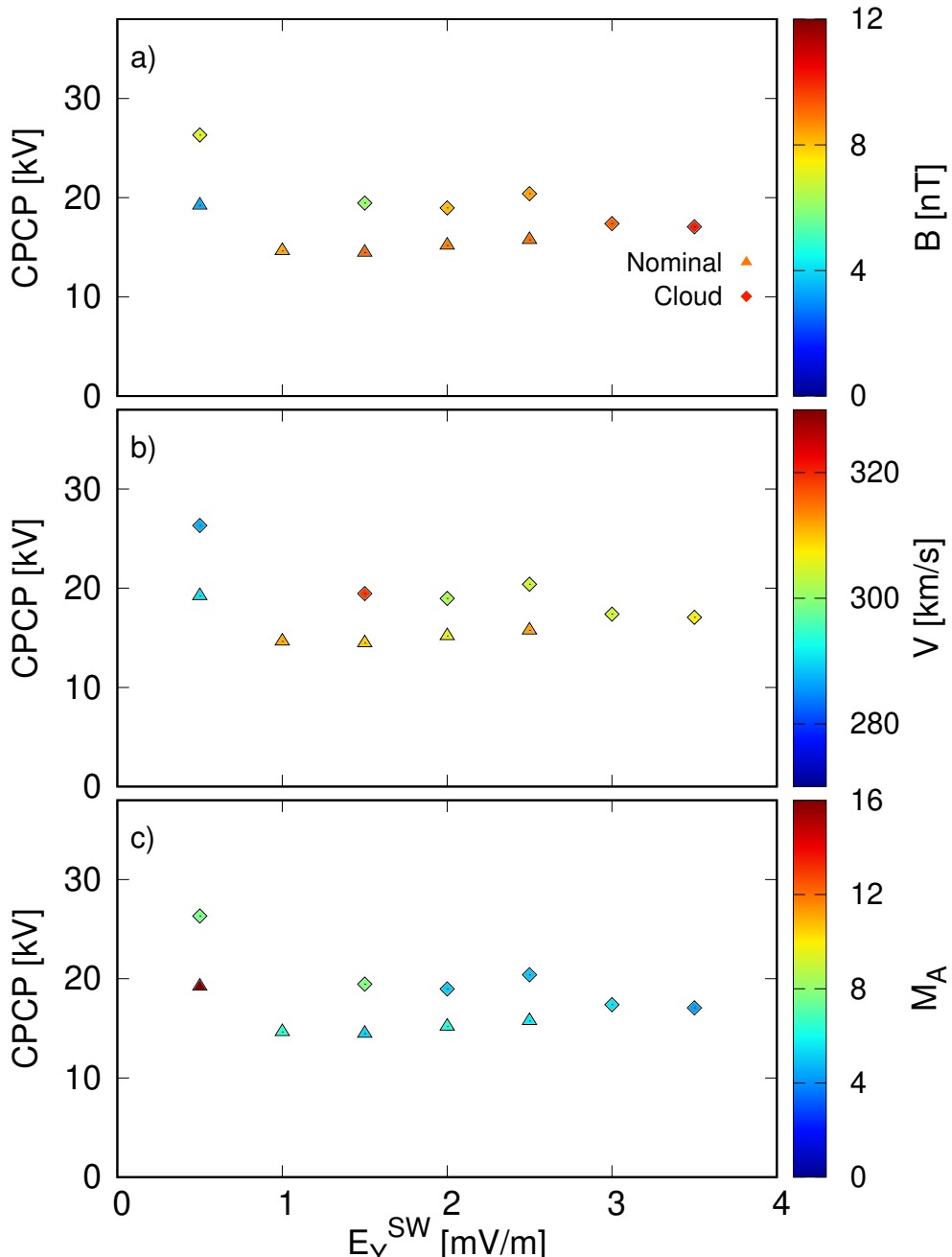

**Figure 9.** The cross-polar cap potential (CPCP) as a function of the IMF $E_Y$ for the 2014 ICME cloud period, with nominal solar wind conditions before and after the ICME event taken into account separately. GUMICS-4 simulation data with 1 minute time resolution has been averaged by 10 minutes and binned by upstream $E_Y$ with 0.5 mV/m intervals. Panels a, b and c show the magnitudes of the IMF, the upstream flow speed and the Alfvén Mach number, respectively.

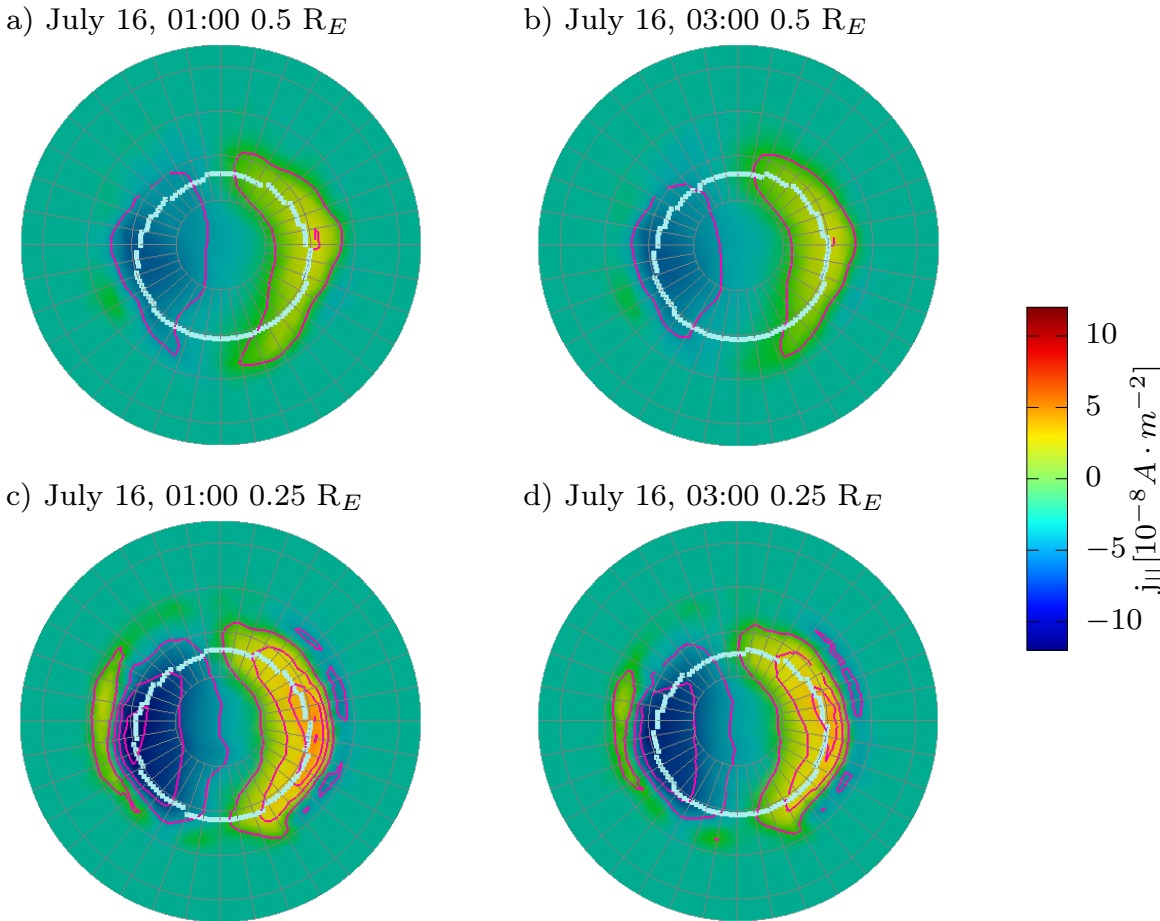

a) July 16, 01:00 0.5 $R_E$

b) July 16, 03:00 0.5 $R_E$

c) July 16, 01:00 0.25 $R_E$

d) July 16, 03:00 0.25 $R_E$

**Figure 10.** The northern hemisphere field-aligned current pattern in GUMICS-4 simulation at 01:00 UT (panels a and c) and at 03:00 UT (panels b and d) in July 16, 2012. Panels a and b (c and d) show the results of the simulation run in which 0.5 (0.25) $R_E$ maximum spatial resolution was used.

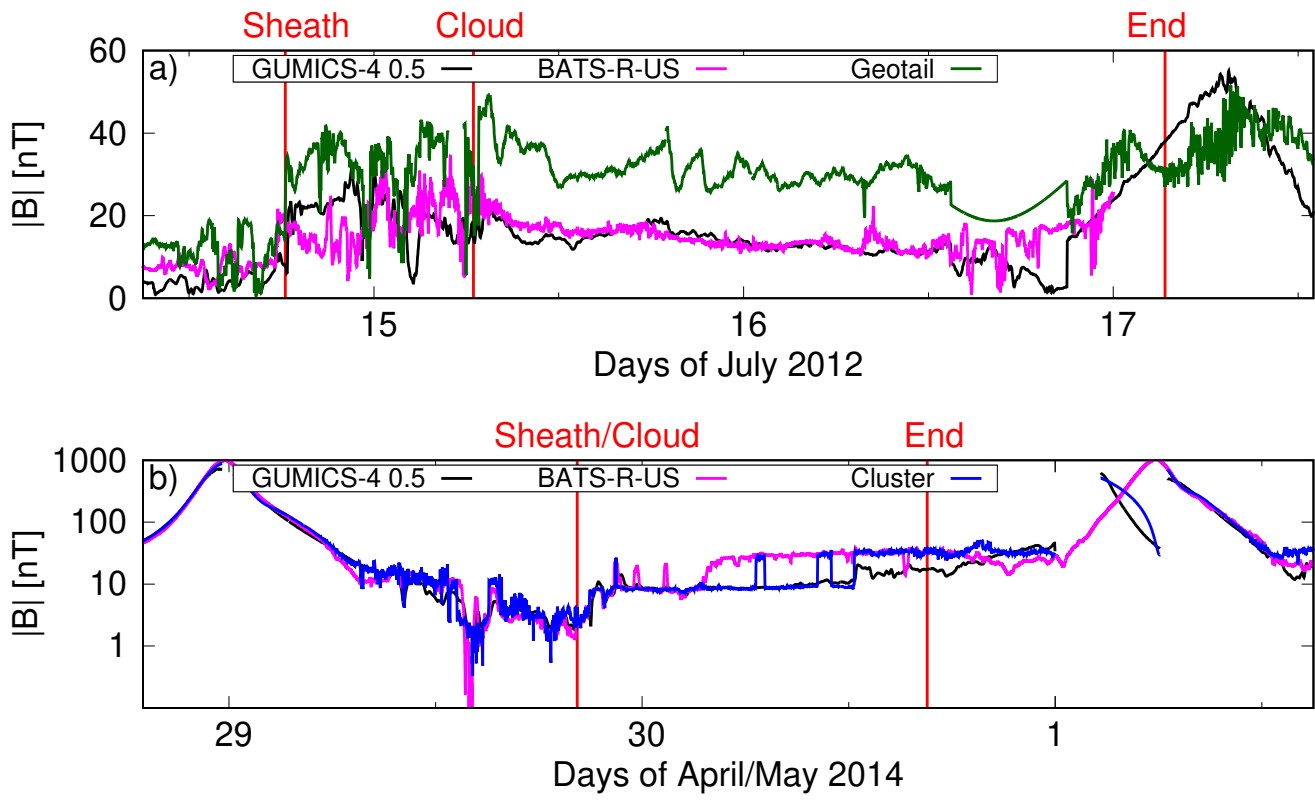

**Figure 11.** The time series of the magnetic field magnitude $|B|$ along the orbits of Geotail during July 14 09:00 UT – July 17 15:00 UT, 2012 (panel a) and Cluster 1 during April 28 19:00 UT – May 1 17:00 UT, 2014 (panel b) as measured by Geotail (green) and Cluster 1 (blue) and predicted by GUMICS-4 (black) and BATS-R-US (magenta).