# Peer review of "GUMICS-4 analysis of ICME impact at Earth during low and typical Mach number solar wind"

_Annales Geophysicae, 2018_

## Referee Comment (RC1) · Anonymous Referee #1 · 25 Aug 2018

Reviewer report on paper "ICME impact at Earth with low and typical Mach number plasma characteristics" by Antti Lakka et al

The authors present their analysis of global MHD (GUMICS model) simulations of solar wind-magnetosphere-ionosphere system during two interplanetary cloud events. They compute the magnetopause standoff distance and magnetic fields along the trajectories of magnetospheric spacecraft (and compare them to empirical models and spacecraft observations), estimate approximately the amount of energy transferred into the magnetosphere, and analyse the potential drop applied in the ionosphere (CPCP) which characterizes the intensity of global convection.

The main problem for me with this paper is that it actually tries to adress two related problems, physical effects (ICME impact differences, e.g. saturation) and technical as-

pects (validation of GUMICS computation results). Second aspect is crucial because, if the computed values are wrong and do not characterize the reality, they can not be used to study the physics in the magnetospheric system. Unfortunately in the paper only the first problem is formally claimed as a paper goal: all Introduction, the paper title and most of the abstract are about the properties of ICME. As concerns the results - the only ICME-related conclusion (last line in the Abstract) is that 'CPCP saturation is affected by the upstream conditions, with strong dependence on the Alfvén Mach number' . In such formulation this is actually well-known from many previous studies, including simulations. So - no new results??

My impression is that throughout the paper the authors are under a strong pressure of technical aspects because the GUMICS validation results are not very optimistic: the B-field comparisons demonstrate big differences between the predictions and observations (whose origin is not identified); the computed CPCP values are much lower than usual ones; their values differ significantly between two simulation runs at standard and doubled resolution, and there is no confidence that the high-resolution run reached the optimum (CPCP values are still low to my view). No clear conclusions about validation success were done in the discussion/conclusion sections.

A big general problem with GMHD simulations is that, for the same solar wind inputs, different GMHD models (their runs at comparable resolution) provide very different answers for essential output parameters (incl.global parameters) –see Gordeev et al. (Space Weather 2015, 2017). For some parameters like the MP standoff distance, the deviations between models were not large. For some other parameters like CPCP and total field-aligned current, their values differ greatly between models, with GUMICS showed too low values of both global variables. This problem is not distinctly articulated in the paper, although a common need of truly global and accurate simulation models justifies paying attention to the technical (validation) aspects as well.

In view of these problems, I believe, the paper in the existing form can not be recommended to the publication. However, I believe, the authors still have potentially interesting material in hands and can possibly find a proper balance between two (physical and technical) aspects to reorganize the manuscript, to clearly formulate and answer the main questions to be addressed, and to expose the new results as a response to the formulated goals (not necessarily being all positive?????).

Some specific comments are given below. ==================================================================
Abstract: it mostly explains what you did, only 1 line (of 11 lines) tells what you obtained (your results).

p.3 - l.3 to7: Paper goal is not actually explained, you only tell that you do simulations during two ICME events and compute magnetopause, but not – which problem are you focusing on in that paper? What drives your choice of computed characteristics (MP distance, energy input, magnetic fields, ..., how it helps to reach the goal?

Figs.1,2: The energetic particle fluxes are not used in the study??? Why don't use logarithmic scale for MA, otherwise the values in the most interesting small MA region are not readable from the plot

p.5, comparison of magnetic fields in the magnetosphere. First, it would be natural to place comparison of simulated/observed fields at the end of this paragraph where you show the results, otherwise (as it is now) the discussion of comparisons (now placed in sect.4.3) stays couple pages later from the corresponding figure, very hard to read.

p.6-26:" Total energy through the dayside magnetopause is computed by evaluating the Poynting flux in the vicinity of the (Shue) magnetopause, and its component parallel to the magnetopause surface normal." Your method to compute the energy flow is not sufficiently introduced and analysed, although there are big questions. The Shue magnetopause stays at some distance from simulated MP, in the region with large spatial gradients of flow and other parameters; also, the shapes of computed and Shue magnetopauses can be different. That means some portions of Shue MP can be in the magnetosheath (with tailward energy flow), some in the magnetosphere(with sunward Poynting flux near the dayside MP). How can you justify your computations? One way

to quickly look on that is to compute energy flows throughout Shue MPs displaced, say by dX = +/-0.2 (or 0.5)Re. Anyway, the uncertainty of such computations should be somehow estimated.

pp.6-7: When validating MP and CPCP it would be reasonable to compare with empirical values for those conditions. I would also recommend to compare your results with Gordeev et al.(2015, doi:10.1002/2015SW001307) validation effort, where the empirical data have been used for testing (e.g., their Fig.9).

Section 4.3.Local dynamics. I think, a so big difference of magnitudes between GUMICS predictions and actual observations in Figs.4,5, a two-fold differences (or more) in many regions, is a kind of bad news for GUMICS validation. However- no analyses is provided – what was wrong in simulated field in these regions? How much the total pressure is wrong? Which components are most affected, etc?? Why don't you show the traces of high-resolution run results on Figs.4,5, are there differences between two runs? I don't see any conclusions from these comparisons, it may not be a good idea to show such bad agreement without explanations.

An interesting aspect: if the saturation works under total FAC being an order of magnitude smaller than real , it may show that magnetospheric mechanisms (e.g. the FAC influence on the dayside magnetospheric magnetic field as suggested by G.Siscoe et al) do not contribute to the saturation effect. This can be a useful side result in case if your high resolution is not yet sufficient to increase CPCP and total FAC toward realistic values.. ———————————————————————————————————-end review

---

## Referee Comment (RC2) · Anonymous Referee #2 · 13 Sep 2018

**1   General Comments**

This paper studies the effect of two ICMEs of different characteristics on the Earth's magnetosphere, focussing on the saturation of the cross polar cap potential (CPCP). The majority of the abstract talks about the properties of the ICMEs, and is lacking in actual results, or the motivation of the paper. The introduction has a good overview of the relevant literature. However, like the abstract, it is missing the aims and motivation of the paper.

Whilst the results are interesting, I have concerns about their validity. The validation performed in the paper is minimal, only comparing simulation to spacecraft magnetic field data and the position of the magnetopause with the Shue model. The comparison

with spacecraft data is missing a key aspect, the plasma data, and there is little explanation for why GUMICS-4 underestimates the magnetic field strength. The comparison of the Shue model with the simulation magnetopause is also missing key details, such as the definition of the "dayside magnetopause" and whether errors include the full 3D simulation magnetopause. A two or three dimensional comparison would be more appropriate.

This leads to the other issue with the paper, the calculation of the total energy into the magnetosphere. The Shue model is an axisymmetric model, and does not include features such as the cusps, hence using the Shue model for this calculation is potentially incorrect, capturing the sheath or magnetosphere.

The overall quality of the writing in the paper is adequate with a few spelling, grammatical and citation style errors. These have been pointed out in the specific and technical comments, though the authors should thoroughly proof read.

Though the results are interesting, I would not recommend the paper for publication in its current form. However, with a little more analysis and responding to the questions posed in this review, it has the potential for publication.

**2  Specific Comments**

- Pg 3, Section 2.1: Do you consider a dipole tilt or rotation? This should be stated

- Pg 4, Ln 14: You should be specific in why it's not feasible. Does it run too slowly, or are there memory issues?

- Pg 4, Ln 11: Should list the solar wind values you're referencing to make it easier to understand

- Pg 6, Ln 26: Why do you use the Shue magnetopause for this calculation, not the

simulation magnetopause? I would have thought this would be a more consistent calculation with the simulation. The general 3D structure of the Shue magnetopause likely not in the correct position, especially near the cusps. Does this mean you'd be capturing energy flux through an arbitrary surface either in the sheath or inside the magnetopause? Also, does this use the 3D magnetopause surface and how far does the dayside region extend to? The details of this calculation should be more clearly stated in the paper (or cited).

- Pg 7, Lns 1-5: Continuing on from the previous comment, are these percentages over the whole 3D dayside surface of the magnetopause? If they aren't then they probably aren't a good metric as they don't account for the full shape of the magnetopause.

- Pg 7, Ln 10: The author mentions both runs are consistent; this should be shown with a figure of the GUMICS magnetosphere data (e.g. cuts through the noon-midnight and ecliptic planes).

- Fig 4c: What is the strange artefact in the position of Geotail? It seems to jump to a different position?

- Fig. 5: More odd artefacts: (c) position of geotail jumps throughout dataset; (b) jumps in the magnetic field strength of Cluster at approx. April 30 (06:00) and April 30 (09:00)

**3   Technical corrections**

- Pg 4, Ln 10: Need brackets around Lakka et al. (2017)

- Pg 5 Ln 6: "rotate", not "rotated"

- Pg 5, Ln 30: Citation should have parentheses

- Pg 6, Ln 11: replace "proper" with "properly"

- Fig 8. Ln 4: replace "are showing" with "show"

- Pg 10: Ln 2: unnecessary hyphen in front of "line"

- Pg 10, Ln 8: Citation should not have parentheses

- Pg 10, Ln 20: Citation should have parentheses

- Pg 11, Ln 14: Citation should have parentheses

---

## Author Comment (AC1) · 13 Oct 2018

**Report #1**

**Author general comment**
*We would like to thank the reviewer for reviewing manuscript "ICME impact at Earth with low and typical Mach number plasma characteristics" and thus helping to improve it. We considered carefully every comment made by the reviewer and prepared responses accordingly. Please find our responses to the comments below.*
* * *
**General comments**

Reviewer report on paper "ICME impact at Earth with low and typical Mach number plasma characteristics" by Antti Lakka et al. The authors present their analysis of global MHD (GUMICS model) simulations of solar wind-magnetosphere-ionosphere system during two interplanetary cloud events. They compute the magnetopause standoff distance and magnetic fields along the trajectories of magnetospheric spacecraft (and compare them to empirical models and spacecraft observations), estimate approximately the amount of energy transferred into the magnetosphere, and analyse the potential drop applied in the ionosphere (CPCP) which characterizes the intensity of global convection.

The main problem for me with this paper is that it actually tries to adress two related problems, physical effects (ICME impact differences, e.g. saturation) and technical aspects (validation of GUMICS computation results). Second aspect is crucial because, if the computed values are wrong and do not characterize the reality, they can not be used to study the physics in the magnetospheric system. Unfortunately in the paper only the first problem is formally claimed as a paper goal: all Introduction, the paper title and most of the abstract are about the properties of ICME. As concerns the results - the only ICME-related conclusion (last line in the Abstract) is that 'CPCP saturation is affected by the upstream conditions, with strong dependence on the Alfvén Mach number'. In such formulation this is actually well-known from many previous studies, including simulations. So - no new results??

> *We thank the reviewer for this comment. It forced us to rethink what we want to say in the paper and obtained new results. The leading thought of the paper is to 1) consider two different ICME events and observe if they produce different effects on the magnetospheric physics by considering several parameters and 2) assess how GUMICS-4 reproduces those events by providing an uncertainty estimate with every parameter. We e.g. show that the accuracy of GUMICS-4 results is dependent on the magnetospheric region under inspection. We have now improved the exposure of the technical aspects starting from abstract and introduction (see pages 1 and 3).*

My impression is that throughout the paper the authors are under a strong pressure of technical aspects because the GUMICS validation results are not very optimistic: the B-field comparisons demonstrate big differences between the predictions and observations (whose origin is not identified); the computed CPCP values are much lower than usual ones; their values differ significantly between two simulation runs at standard and doubled resolution, and there is no confidence that the high-resolution run reached the optimum (CPCP values are still low to my view). No clear conclusions about validation success were done in the discussion/conclusion sections.

*We agree with the reviewer in the sense that GUMICS-4 produces different results when compared with e.g. in-situ satellite observations or measured polar cap potential. It is not even a surprise, since there have been many studies before reporting how well GUMICS-4 (or any other global MHD code) captures the magnetospheric dynamics. The problem is partly due to MHD physics not being sufficient, but also partly because the compared quantities may not represent the same quantities at all. For instance, a corresponding observation for global MHD CPCP is hard to find. Some studies have used PCN index, which doesn't really represent a global CPCP value. Others have used potentials deduced from ionospheric radars, but still do not capture the entire polar cap area. Given these difficulties in the validation, our best approach is to use well-known references, validate simulation results, report the shortcomings of our validation, and assess how well we succeeded. This is just what we do in our paper and we hope that it is now easier to see especially in the discussion and conclusion sections of the revised manuscript.*

A big general problem with GMHD simulations is that, for the same solar wind inputs, different GMHD models (their runs at comparable resolution) provide very different answers for essential output parameters (incl.global parameters) –see Gordeev et al. (Space Weather 2015, 2017). For some parameters like the MP standoff distance, the deviations between models were not large. For some other parameters like CPCP and total field-aligned current, their values differ greatly between models, with GUMICS showed too low values of both global variables. This problem is not distinctly articulated in the paper, although a common need of truly global and accurate simulation models justifies paying attention to the technical (validation) aspects as well.

*Different GMHD models have different strengths. Deviations in e.g. CPCP values are caused by differences in how the models handle excessive amount of electric current through the polar cap, which causes some models to underestimate, others to overestimate CPCP. Using GMHD model requires knowledge of the general features of the model performance and understanding their strengths and limitations. Sheding light to this issue is one of the key targets of this paper. From our paper point of view, comparing the time evolution of e.g. CPCP between GUMICS-4 and reference parameter is important. This aspect is articulated better in the revised manuscript.*

In view of these problems, I believe, the paper in the existing form can not be recommended to the publication. However, I believe, the authors still have potentially interesting material in hands and can possibly find a proper balance between two (physical and technical) aspects to reorganize the manuscript, to clearly formulate and answer the main questions to be addressed, and to expose the new results as a response to the formulated goals (not necessarily being all positive?????).
* * *
**Specific comments**
* * *
p.3 - l.3 to7: Paper goal is not actually explained, you only tell that you do simulations during two ICME events and compute magnetopause, but not – which problem are you focusing on in that paper? What drives your choice of computed characteristics (MP distance, energy input, magnetic fields, . . ., how it helps to reach the goal?

*We agree with the reviewer. Those parameters are used because they are strongly affected by (especially strong) ICME events. The goal of this paper is to see how the parameters are affected by ICMEs with different strength AND how accurate GUMICS-4 results are in those (ICME) conditions. To achieve our goal, we use those parameters and compare*

*simulation results with known references and compute uncertainty estimate. The end of the introduction section hopefully highlights these issues better now. Please see page 3.*

Figs.1,2: The energetic particle fluxes are not used in the study??? Why don't use logarithmic scale for MA, otherwise the values in the most interesting small MA region are not readable from the plot

> *We thank the reviewer for this comment. We adopted logarithmic scale for MA since it really makes figs 1,2 a lot better. However, even if energetic particle fluxes are not directly used in the study, showing them along with solar wind data provides additional information in a sense that it verifies magnetic cloud onset time especially for the 2012 event; gradual decrease of proton flux is observed at the same time with solar wind density decrease. On the other hand, absence of such flux decrease in 2014 shows that the event truly is moderate compared with the 2012 event.*

p.5, comparison of magnetic fields in the magnetosphere. First, it would be natural to place comparison of simulated/observed fields at the end of this paragraph where you show the results, otherwise (as it is now) the discussion of comparisons (now placed in sect.4.3) stays couple pages later from the corresponding figure, very hard to read.

> *We think that the actual results are better to be found in the same section (Analysis) together with global dynamics results. Otherwise we would have to choose which results we are reporting in section 3 already (just measured Bmag or GUMICS-4 Bmag as well, what about the relative differences shown in the revised figures 4 and 5?)*

p.6-26:" Total energy through the dayside magnetopause is computed by evaluating the Poynting flux in the vicinity of the (Shue) magnetopause, and its component parallel to the magnetopause surface normal." Your method to compute the energy flow is not sufficiently introduced and analysed, although there are big questions. The Shue magnetopause stays at some distance from simulated MP, in the region with large spatial gradients of flow and other parameters; also, the shapes of computed and Shue magnetopauses can be different. That means some portions of Shue MP can be in the magnetosheath (with tailward energy flow), some in the magnetosphere(with sunward Poynting flux near the dayside MP). How can you justify your computations? One way to quickly look on that is to compute energy flows throughout Shue MPs displaced, say by dX = +/-0.2 (or 0.5)Re. Anyway, the uncertainty of such computations should be somehow estimated.

> *We provide detailed explanation of the used method in the revised manuscript. It is true that the shape of the Shue magnetopause probably differs from the actual magnetopause. Previously, Palmroth et. al. (doi:10.1029/2002JA009446) computed the shape of actual magnetopause from GUMICS-4 results and compared energy transfer to the epsilon parameter. They also showed that the energy perpendicular to the boundary did not change with small displacement of the boundary thus demostrating the robustness of the method to calculate the incoming energy. Instead, we consider the Shue magnetopause surface by displacing its nose 30% Sunward. We use 30% since it is maximum relative difference in magnetopause position between GUMICS-4 and the Shue model. This prevents underestimation of the size of the magnetosphere. The method used here gives values for energy of the same order of magnitude compared to study by Palmroth (mentioned above). Thus, we have good confidence in the methodology. See pages 8-9.*

pp.6-7: When validating MP and CPCP it would be reasonable to compare with empirical values for those conditions. I would also recommend to compare your results with Gordeev et al.(2015, doi:10.1002/2015SW001307) validation effort, where the empirical data have been used for testing (e.g., their Fig.9).

> *We agree. In order to be consistent when using references, we have compared MP, energy transfer and CPCP to known references. For MP the reference is the Shue model, for energy tranfer it's the epsilon parameter, and for the CPCP it is PCI (Ridley, Polar cap index comparisons with AMIE cross polar cap potential, electric field, and polar cap area, (2004)) deduced from PNC index. All of these have been used in previous studies and are easy to plot alongside GUMICS-4 results. Comparisons to PCI and epsilon were missing from the previous manuscript version, but are added in the revised version (see figures 6 and 7 and section 4.1). Moreover, we provide a framework to our study by comparing our results to work by Gordeev (see Discussion section).*

Section 4.3.Local dynamics. I think, a so big difference of magnitudes between GUMICS predictions and actual observations in Figs.4,5, a two-fold differences (or more) in many regions, is a kind of bad news for GUMICS validation. However- no analyses is provided – what was wrong in simulated field in these regions? How much the total pressure is wrong? Which components are most affected, etc?? Why don't you show the traces of high-resolution run results on Figs.4,5, are there differences between two runs? I don't see any conclusions from these comparisons, it may not be a good idea to show such bad agreement without explanations.

> *We agree. The discussion of the results in this section is now improved. We show that accuracy of GUMICS-4 is dependent on which part of the magnetosphere is considered. The absolute value of Bmag in GUMICS-4 agrees better when Bmag is high (S/C is close to the Earth).*

An interesting aspect: if the saturation works under total FAC being an order of magnitude smaller than real , it may show that magnetospheric mechanisms (e.g. the FAC influence on the dayside magnetospheric magnetic field as suggested by G.Siscoe et al) do not contribute to the saturation effect. This can be a useful side result in case if your high resolution is not yet sufficient to increase CPCP and total FAC toward realistic values..

> *Thank you for this comment. We considered it carefully in the text.*

---

## Author Comment (AC2) · 13 Oct 2018

**Author general comment**
*We would like to thank the reviewer for reviewing manuscript "ICME impact at Earth with low and typical Mach number plasma characteristics" and thus helping to improve it. We considered carefully every comment made by the reviewer and prepared responses accordingly. Please find our responses to the comments below.*
* * *
**General comments**

This paper studies the effect of two ICMEs of different characteristics on the Earth's magnetosphere, focussing on the saturation of the cross polar cap potential (CPCP). The majority of the abstract talks about the properties of the ICMEs, and is lacking in actual results, or the motivation of the paper. The introduction has a good overview of the relevant literature.

However, like the abstract, it is missing the aims and motivation of the paper. Whilst the results are interesting, I have concerns about their validity. The validation performed in the paper is minimal, only comparing simulation to spacecraft magnetic field data and the position of the magnetopause with the Shue model.

*We thank the reviewer for this comment. The manuscript is revised carefully to increase the amount of validation. Every parameter considered comes now with validation. Based on this, conclusions are also made of the accuracy of GUMICS-4 results.*

The comparison with spacecraft data is missing a key aspect, the plasma data, and there is little explanation for why GUMICS-4 underestimates the magnetic field strength.

*While the plasma motions are critically important, in this case it is not easy due to the fact that the S/C reside most of the time in the low-density lobe regions, where the observations suffer from the very low counts. Furthermore, there are large data gaps in the observations hindering the comparisons.*

The comparison of the Shue model with the simulation magnetopause is also missing key details, such as the definition of the "dayside magnetopause" and whether errors include the full 3D simulation magnetopause. A two or three dimensional comparison would be more appropriate.

*The Shue magnetopause nose position is a single grid point in GUMICS-4 results. This is explained in the revised manuscript. See page 7, line 29. Similarly, the dayside magnetopause is a 3D surface computed from its nose position, extending from 0 RE in Sunward direction. See page 7.*

This leads to the other issue with the paper, the calculation of the total energy into the magnetosphere. The Shue model is an axisymmetric model, and does not include features such as the cusps, hence using the Shue model for this calculation is potentially incorrect, capturing the sheath or magnetosphere.

*When computing the 3D Shue magnetopause for evaluating the amount of transferred energy, we have displaced its nose position by 30% Sunward to avoid inclusion of*

*magnetosphere. The methods are explained in detail in the revised manuscript. Moreover, earlier studies have demonstrated the robustness of the energy computations (Palmroth et. al. (doi:10.1029/2002JA009446)). See pages 8-9.*

The overall quality of the writing in the paper is adequate with a few spelling, grammatical and citation style errors. These have been pointed out in the specific and technical comments, though the authors should thoroughly proof read. Though the results are interesting, I would not recommend the paper for publication in its current form. However, with a little more analysis and responding of the questions posed in this review, it has the potential for publication.
* * *
**Specific comments**
* * *
• Pg 3, Section 2.1: Do you consider a dipole tilt or rotation? This should be stated

> *We agree. Dipole field was rotating and the angle was nonzero. This is explained in the revised manuscript (see page 4).*

• Pg 4, Ln 14: You should be specific in why it's not feasible. Does it run too slowly, or are there memory issues?

> *Simulations would take way too much time, probably months. In the revised manuscript we state that "...not feasible due to long simulation physical time (up to 3.5 days) and resulting long simulation running times." (see page 4).*

• Pg 4, Ln 11: Should list the solar wind values you're referencing to make it easier to understand

> *We agree. The used solar wind values are listed in the revised manuscript. (see page 4).*

• Pg 6, Ln 26: Why do you use the Shue magnetopause for this calculation, not the simulation magnetopause? I would have thought this would be a more consistent calculation with the simulation. The general 3D structure of the Shue magnetopause likely not in the correct position, especially near the cusps. Does this mean you'd be capturing energy flux through an arbitrary surface either in the sheath or inside the magnetopause? Also, does this use the 3D magnetopause surface and how far does the dayside region extend to? The details of this calculation should be more clearly stated in the paper (or cited).

> *We thank the reviewer for this comment. Previously, Palmroth et. al. (doi:10.1029/2002JA009446) computed the shape of actual magnetopause from GUMICS-4 results and compared energy transfer to epsilon parameter. They also showed that the energy perpendicular to the boundary did not change with small displacement of the boundary thus demostrating the robustness of the method to calculate the incoming energy. Instead, we consider the dayside (extends to 0 RE) Shue magnetopause 3D surface by displacing its nose 30% Sunward. We use 30% since it is maximum relative difference in magnetopause position between GUMICS-4 and the Shue model. This prevents underestimation of the size of the magnetosphere. The method used here gives values for energy of the same order of magnitude compared to study by Palmroth (mentioned above). Thus, we have good confidence in the methodology. We provide detailed explanation of the used method in the revised manuscript. See pages 8-9.*

• Pg 7, Lns 1-5: Continuing on from the previous comment, are these percentages over the whole 3D dayside surface of the magnetopause? If they aren't then they probably aren't a good metric as they don't account for the full shape of the magnetopause.

> ***The percentages are for the nose position. As the 3D Shue magnetopause structure is characterized by the position of the nose, we (and several other authors whose work we cite) strongly believe that this gives a good overview of the accuracy of the GUMICS-4 magnetopause position. Different models (empirical and simulations) produce different flaring in the distant magnetotail, but mostly agree within the dayside and near-tail region. Thus, considering the entire magnetopause Sunward of -30 RE would lead to similar conclusions. We do mention several times in the text that we only consider magnetopause nose to make that point clear.***

• Pg 7, Ln 10: The author mentions both runs are consistent; this should be shown with a figure of the GUMICS magnetosphere data (e.g. cuts through the noon-midnight and ecliptic planes).

> ***We agree. However, as section 4.1 is revised, the paragraph in question is not relevan anymore and has been removed.***

• Fig 4c: What is the strange artefact in the position of Geotail? It seems to jump to a different position?

> ***The artefacts are errors made when interpolating SC location over data gaps. These errors are removed from the revised figure 4.***

• Fig. 5: More odd artefacts: (c) position of geotail jumps throughout dataset; (b) jumps in the magnetic field strength of Cluster at approx. April 30 (06:00) and April 30 (09:00)

> ***Regarding (c): Also interpolation error, which does not show in the revised version of figure 5. Regarding (b): The measured Bmag increases from approx. 10 nT to 30 nT, which is on par with the increase of measured Bmag at 12:00 (April 30). At same time the magnetopause is in motion toward the Earth, and the S/C resides close to it, which probably explains those rapid increases.***
* * *
**Technical comments**

• Pg 4, Ln 10: Need brackets around Lakka et al. (2017)

> ***Corrected in the revised manuscript. See page 4.***

• Pg 5 Ln 6: "rotate", not "rotated"

> ***Corrected in the revised manuscript. See page 5.***

• Pg 5, Ln 30: Citation should have parentheses

> ***Corrected in the revised manuscript. See page 6.***

• Pg 6, Ln 11: replace "proper" with "properly"

*Word "proper" is referring to the "actual" part of the magnetic cloud, and that's why we think that its usage in the context of the text is justifiable.*

• Fig 8. Ln 4: replace "are showing" with "show"

*Corrected in the revised manuscript. See figure 8 caption.*

• Pg 10: Ln 2: unnecessary hyphen in front of "line"

*Corrected in the revised manuscript. See page 11.*

• Pg 10, Ln 8: Citation should not have parentheses

*Corrected in the revised manuscript. See page 11.*

• Pg 10, Ln 20: Citation should have parentheses

*Corrected in the revised manuscript. See page 12.*

• Pg 11, Ln 14: Citation should have parentheses

*Corrected in the revised manuscript. See page 13.*

---

## Referee Report (RR1)

Reviewer report on paper "ICME impact at Earth with low and typical Mach number plasma characteristics" by Antti Lakka et al

The authors invested efforts to improve the manuscript by addressing the referee's comments. They reformulated motivation, added the GUMICS validation as one of two main goals, added couple more reference parameters (PCI and ε)for validation purposes, computed the deviations between predictions and  (now in Table 3-6) and extended the discussion (paper expanded a lot and now the text is on 17 pages as compared to previous 12pages).
Still I have to reiterate two previous conclusions. One is that the paper does not expose any new result related to ICME effects on magnetospheric dynamics (one of two goals of the paper).
Another one is related to the additional statement (which appeared in both abstract and Conclusion section), that "GUMICS-4 results are in a good agreement with the reference values."
I can not accept this statement, because the materials presented rather show the opposite things.

(1) In fact, statistical evaluation gives good marks (deviation of the order of 5%) only for standoff distance parameter (which, by its physics, is mostly dictated by SW flow pressure, has rather small relative variations, and all MHD models agreeing rather well with predictions in the large range of conditions, as was also showed in Gordeev et al.SW 2015 paper). This is not much surprising and quite expected, even the simple Chapman-Ferraro model with pressure balance  provides quite successful predictions of the subsolar nose distance variations.

(2) The validation success critically depends on the choice of 'reference parameters', intended to show the realistic values of key global characteristics of the system to be compared with model predictions.  I was surprized by your choice of PCI index as a reference to cross-polar cap potential drop (CPCP),  which is actually an indirect uncalibrated proxy (based on PC index value) with unknown (and hardly good) accuracy. Why not using the potential model by Weimer et al (JGR 2005), based on direct E-field integration results from thousands of direct DE2 polar cap crossings occurred under various SW conditions and parametrized dy SW parameters? Or some other data-based representative proxy?? Anyway they will be more representative compared to your magnetogram-based proxy.
In the same way I hardly can take seriously the ε-parameter as a realistic reference to actual energy consumption, even although it is used as such in a number of studies.
As you also mention in the text, that these indices are very indirect proxy, but still continue to use them in quantitative comparisons. I believe the material in Table 3 you completeand corresponding discussions is unrelevant.  Anyway it also shows rather bad agreement (relative difference of 30 to 70% for average values).

(3) Using B-field measurements in different parts of the system is a good (although difficult) choice.
By some reasons you compare only difference of B-magnitudes (magntitude of vector difference will be larger), but even with this choice you systematically infer very large deviations (40 to 80% in Table 4, 35 to 60% in Table 5, using different averaging rules), indicating that GUMICS predicted significantly smaller magnetic fields everywhere in the system than those which Geotail and Cluster actually measured.  For me this would be an indication of a kind of disaster in the code performance.
This can not be explained by a manifestaton of local mismatch between predictions and observations (which can sometimes affect the temporal variations or similar), because the effect is a system-wide one. The origin of mismatch is NOT discussed and it is not investigated in the paper.

SUMMARY: Big mismatch in predicted/observed B-field and also between CPCP values seem to indicate serious problems in the GUMICS performance. (Good consistence for magnetopause stand-off distance is of little consolation because it is well predicted by all models, starting from Chapman-Ferraro model.)
I would rather reformulate your statement as "GUMICS-4 results are in a serious disagreement with the reference values."
In this situation it seems that there are no sufficient reasons to suggest the publication of this paper

----------------------------------------------------------------------------end review

---

## Author Response (AR2)

**Report #1**

**Author general comment**
*We would like to thank the reviewer for reviewing manuscript "ICME impact at Earth with low and typical Mach number plasma characteristics" and thus helping to improve it. We considered carefully every comment made by the reviewer and prepared responses accordingly. Please find our responses to the comments below.*
* * *
**General comments**

The authors invested efforts to improve the manuscript by addressing the referee's comments. They reformulated motivation, added the GUMICS validation as one of two main goals, added couple more reference parameters (PCI and epsilon) for validation purposes, computed the deviations between predictions and (now in Table 3-6) and extended the discussion (paper expanded a lot and now the text is on 17 pages as compared to previous 12pages). Still I have to reiterate two previous conclusions. One is that the paper does not expose any new result related to ICME effects on magnetospheric dynamics (one of two goals of the paper).

**Author comment**
*We disagree with this comment. Our paper shows that the saturation of the polar cap potential takes place using "real" solar wid data and relatively coarse spatial grid. In addition, we even suggest that the field-aligned current pattern may not be the decisive factor in the saturation effect. We feel that since there is no concensus wehere the stauration effect takes place after decades of studies, we are contributing on the topic by showing that it may not take place in the inner magnetosphere.*

Another one is related to the additional statement (which appeared in both abstract and Conclusion section), that "GUMICS-4 results are in a good agreement with the reference values."
I can not accept this statement, because the materials presented rather show the opposite things.

**Author comment**
*Thank you for this comment. This statement is reformulated in the revised manuscript (see p.1 l.11 and p.17 l.13)*

(1) In fact, statistical evaluation gives good marks (deviation of the order of 5%) only for standoff distance parameter (which, by its physics, is mostly dictated by SW flow pressure, has rather small relative variations, and all MHD models agreeing rather well with predictions in the large range of conditions, as was also showed in Gordeev et al.SW 2015 paper). This is not much surprising and quite expected, even the simple Chapman-Ferraro model with pressure balance provides quite successful predictions of the subsolar nose distance variations.

**Author comment**
*We agree with the reviewer. However, it is crucial for the paper to show that our simulation runs are valid in a sense that the deviations in the magnetopause position are small, because if they were large, it would be pointless to use other metrics, since obviously there would be something wrong with the runs. Note that we are trying to contribute on establishing a reasonable choice of reference parameters for future use. That was also one of the goals of this paper.*

(2) The validation success critically depends on the choice of 'reference parameters', intended to show the realistic values of key global characteristics of the system to be compared with model predictions. I was surprised by your choice of PCI index as a reference to cross-polar cap potential drop (CPCP), which is actually an indirect uncalibrated proxy (based on PC index value) with unknown (and hardly good) accuracy. Why not using the potential model by Weimer et al (JGR 2005), based on direct E-field integration results from thousands of direct DE2 polar cap crossings occurred under various SW conditions and parametrized dy SW parameters? Or some other data-based representative proxy?? Anyway they will be more representative compared to your magnetogram-based proxy. In the same way I hardly can take seriously the epsilon-parameter as a realistic reference to actual energy consumption, even although it is used as such in a number of studies. As you also mention in the text, that these indices are very indirect proxy, but still continue to use them in quantitative comparisons. I believe the material in Table 3 you completeand corresponding discussions is unrelevant. Anyway it also shows rather bad agreement (relative difference of 30 to 70% for average values).

> **Author comment**
> *We agree with the reviewer that it is difficult to make use of the two parameters (PCI and epsilon) as validation metrics when considering global MHD codes. However, they are commonly used in the previous papers and we wanted to have some degree of consistency with them. We would also like to note that the use of relative difference is restricted to the validation of the magnetopause stand-off distance in the revised manuscript. However, computed standard deviations (SD) give reasonable results especially for the epsilon parameter (0.725, very close to the magnetopause SD) in 2014. By comparing it to the corresponding 2012 value (2.263) and the ones obtained for the PCI (15.838 in 2012 and 5.107 in 2014) we can conclude that the epsilon parameter provides better comparison (between GUMICS and the reference) of the two and that GUMICS is closer to the references in 2014 (moderate solar wind driving), a trend that is shown throughout the paper.*

(3) Using B-field measurements in different parts of the system is a good (although difficult) choice. By some reasons you compare only difference of B-magnitudes (magntitude of vector difference will be larger), but even with this choice you systematically infer very large deviations (40 to 80% in Table 4, 35 to 60% in Table 5, using different averaging rules), indicating that GUMICS predicted significantly smaller magnetic fields everywhere in the system than those which Geotail and Cluster actually measured. For me this would be an indication of a kind of disaster in the code performance. This can not be explained by a manifestaton of local mismatch between predictions and observations (which can sometimes affect the temporal variations or similar), because the effect is a system-wide one. The origin of mismatch is NOT discussed and it is not investigated in the paper.

> **Author comment**
> *Thank you for pointing this out. We chose the magnitude of B for a reason. It reveals something that definitely should be considered in future studies: The deviation between predicted (by a GMHD model) and measured Bmag can be quite large (up to 80%, as was pointed out by you) during a CME, especially during a relatively effective one, like the one that occurred in July 2012. What our paper failed to do however is to show that this is not something related to GUMICS. In the revised manuscript we show (see figure 11 and p.15 l.31 – p16 l.22) that BATS-R-US code reproduces similar deviations as well. In fact, during the 2014 CME magnetic cloud out of the two models it is GUMICS that is mostly closer to the Cluster measurements. In these conditions we can't help but conclude that modelling (geo)effective CMEs affects the magnetosphere in a global MHD code such that Bmag deviates significantly from in-situ data. What is actually the reason is out of*

*the scope of the current paper, as we are only trying to validate our metrics. We would have probably missed this point had we considered only vector components of B.*
* * *
**Report #2**

**Author general comment**
*We would like to thank the editor for making an effort and reviewing manuscript "ICME impact at Earth with low and typical Mach number plasma characteristics". We considered these comments and made our best answering to them.*
* * *
**General comments**

Dear Dr. Lakka,

thank you for submitting the revision of your manuscript "ICME impact at Earth with low and typical Mach number plasma characteristics". As your manuscript required major revisions, I have sent it back to the two original reviewers, but unfortunately got an answer only from one.

As you will see, the reviewer is of the opinion that while there is clear improvement on the presentation aspects of the work, he/she expresses strong concerns about the originality and the validity of your simulation results. In particular, he/she cites large differences between the measured and simulated parameters (e.g. 40-80% in the magnetic field magnitude), the origin of which is not discussed, while there are some concerns about the methodology used for data/simulations comparisons (i.e. the reference parameters used).

**Author comment**
*We agree that the differences are large. As for the concerns about the comparison procedure, we have restricted the use of the relative difference to the validation of the magnetopause stand-off distance in the revised manuscript. We are also providing discussion on the large deviations in Bmag in GUMICS-4 and show that BATS-R-US code shows similar deviations as well (see figure 11 and p.15 l.31 − p16 l.22).*

I did my own literature search in order to understand what are the typical levels of mismatch between Global MHD simulations and data. The most helpful study in that respect is by Ridley et al. (2016): "Rating global magnetosphere model simulations through statistical data-model comparisons", where GUMICS is also one the codes tested. This statistical study shows differences in the magnetic field components that are typically below 20-30% for Bz and even though differences can be larger in Bx and By, an overall mismatch of 40-80% in your simulations indeed appears large. Smaller differences are also shown in another GUMICS based study by Facsko et al. (2015), "One year in the Earth's magnetosphere: A global MHD simulation and spacecraft measurements".

**Author comment**
*We agree that our results are different that what was achieved by Ridley et al. However, it should be noted that out of the 662 CCMC simulation runs considered in that paper, GUMICS was used in only 12 of them. Also, in a GUMICS Bmag vs. in-situ Bmag comparison provided by Facsko et al. solar wind upstream conditions were quite nominal, at least no CMEs were observed. Thus it could be argued that comparing those studies*

*with our study is like comparing apples and oranges. Nonetheless, we are citing both papers in the revised manuscript as we think that those large deviations that we got in our study are caused by the upstream conditions rather than GUMICS-4.*

However, because my understanding is that you argue about correlated variations between GUMICS simulations and measurements I would like to offer your the opportunity to answer to the referee comments and revise your manuscript.

I strongly recommend to add any relevant discussion points, clarifications and plots that may cover any concerns expressed by Reviewer #1, unless you can justify why there is no need to follow those recommendations:

a) Comparison between magnetic field components, rather than just the magnetic field magnitude
b) Discussion on the choice of parameter for the GUMICS/data comparisons
c) Discussion on the large absolute value deviations between GUMICS and data.

> **Author comment**
> *As pointed out earlier, we show that BATS-R-US code reproduces Bmag (and the large deviations) pretty much as GUMICS, and thus the reason for the deviations can't be GUMICS. By plotting only components of the magnetic field we would have probably lost this very important information, which is telling a message that is in agreement with the rest of the paper: Comparison metrics should be chosen cautiously even for Bmag, which is used extensively as a simulation validation metrics in the previous papers, just like all the other comparison parameters that we chose. We wanted to preserve some level of consistency with earlier works, and thus those parameters were chosen. However, we think that in order to avoid large deviations, comparison results should be interpreted with great care, and that's why we are mostly using the standard deviations as validation method.*

[revised manuscript text omitted]

---

## Author Response (AR3)

**Author general comment**
*We would like to thank the reviewer for reviewing manuscript "ICME impact at Earth with low and typical Mach number plasma characteristics" and thus helping to improve it. We considered carefully every comment made by the reviewer and prepared responses accordingly. Please note that we have uploaded a supplementary figure. Please find our responses to the comments below.*
* * *
**Minor comments**

The authors invested efforts to respond to critical comments and, particularly, added a comparison to the simulation results from another MHD code (BATSRUS, also made with not best resolution). The paper fixes a state-of-art with GMHD modeling of extreme events made with some particular codes at medium resolution.

**Author comment**
*Improving spatial resolution didn't lead to significant differences in our study (see figures 4 and 5), and as BATSRUS and GUMICS-4 share similar features as a result of information exchange during the development of both codes (see e.g. Janhunen et al. 2012: The GUMICS-4 global MHD magnetosphere–ionosphere coupling simulation), it is therefore reasonable to assume that enhancing resolution in BATSRUS would result as better agreement with in-situ data especially because the resolution in BATS-R-US runs is generally better than in GUMICS-4. See p.16 l.28-30.*

I have no objection as regards the publication of current version of paper after minor changes.
I add a few brief notes which may be of some help in the following work on paper/on the subject.
The paper title is about 'ICME impact properties' whereas actually most of paper is about – what the MHD code show in such extreme situation; I would suggest to use smth like "ICME characteristics as computed in global MHD simulation." to orient the reader properly.

**Author comment**
*We have changed the title to "GUMICS-4 analysis of ICME impact at Earth during low and typical Mach number solar wind".*

The abstract seems to have no end, needs some concluding sentence (?)

**Author comment**
*We agree. The abstract is now revised accordingly. See p.1.*

p.6,l.11: "Typically, GUMICS-4 CPCP values are slightly lower than the observed values (Gordeev et al., 2015)."
Slightly?? According to Fig.9 of Gordeev et 2015 paper, GUMICS results for CPCP in average are ~6-8 times lower than empirical values computed from Weimer model for similar solar wind conditions.

**Author comment**
*Agreed, however Gordeev et al. Used low grid resolution in GUMICS-4. The difference between observations and GUMICS-4 lowers significantly when higher resolution is used, see figure 7. This needs to be clarified in the text as well, and thus the text is reformulated in page 6, see p.6 l.11-14.*

Same level of agreement between GUMICS and BATSRUS for july 2012 ICME – is not that encouraging. (Rather we can say that both codes work similarly bad. It is however instructive, that rather good marks can be obtained for strong ICMEs by BATSRUS+RCM combination – see e.g. recent paper DOI: 10.1029/2019SW002157., Testing efficiency of empirical, adaptive, and global MHD magnetospheric models to represent the geomagnetic field in a variety of conditions," by Kubyshkina et al.).

**Author comment**
*Interesting results indeed. We are referring to this paper in the revised manuscript (see p.16 l.33. Something that is common for our paper and paper by Kubyshkina et al. is that GMHD model and empirical model(s) agree reasonably well, however in our paper both show larger error when compared with in-situ measurements (see the supplementary figure). As all the evidence is suggesting that something in the events studied by us (especially the 2012 event) and by Kubyshkina et al. are causing these errors of different magnitude, it would be interesting elaborate on this. However this is beyond the scope of the current paper.*

The questions – Why agreement is so bad in Cluster orbits?? What is missed in the GMHD current system that it shows bad agreement with magnetic observations ? -are not addressed et al.

**Author comment**
*We think that the problem does not concern only GMHD. As you can see from the figure we uploaded as supplementary material, Tsyganenko model reproduces similar difference to the Cluster observations. It thus remains to conclude that the 2012 ICME produces unusually strong compression, which leads to larger field than predicted by either MHD (GUMICS-4 and BATS-R-US) or empirical Tsyganenko model. Note that the particular ICME was one of the strongest of 2012 by the mean magnetic field magnitude value during magnetic cloud. It is however needed to add a few lines to sum up this in the text too. See p.16 l.23-35.*

[revised manuscript text omitted]